# Structural snapshots of La Crosse virus polymerase reveal the mechanisms underlying *Peribunyaviridae* replication and transcription

Benoît Arragain [1,6], Quentin Durieux Trouilleton [1], Florence Baudin [2], Jan Provaznik [3], Nayara Azevedo[3], Stephen Cusack [4✉], Guy Schoehn [1] & Hélène Malet [1,5✉]

Segmented negative-strand RNA bunyaviruses encode a multi-functional polymerase that performs genome replication and transcription. Here, we establish conditions for in vitro activity of La Crosse virus polymerase and visualize its conformational dynamics by cryo-electron microscopy, unveiling the precise molecular mechanics underlying its essential activities. We find that replication initiation is coupled to distal duplex promoter formation, endonuclease movement, prime-and-realign loop extension and closure of the polymerase core that direct the template towards the active site. Transcription initiation depends on C-terminal region closure and endonuclease movements that prompt primer cleavage prior to primer entry in the active site. Product realignment after priming, observed in replication and transcription, is triggered by the prime-and-realign loop. Switch to elongation results in polymerase reorganization and core region opening to facilitate template-product duplex formation in the active site cavity. The uncovered detailed mechanics should be helpful for the future design of antivirals counteracting bunyaviral life threatening pathogens.

[1] Univ. Grenoble Alpes, CNRS, CEA, IBS, F-38000 Grenoble, France. [2] European Molecular Biology Laboratory (EMBL), Structural and Computational Biology Unit, 69117 Heidelberg, Germany. [3] European Molecular Biology Laboratory (EMBL), GeneCore, 69117 Heidelberg, Germany. [4] European Molecular Biology Laboratory (EMBL), 38000 Grenoble, France. [5] Institut Universitaire de France (IUF), Paris, France. [6] Present address: European Molecular Biology Laboratory (EMBL), 38000 Grenoble, France. ✉email: cusack@embl.fr; helene.malet@ibs.fr

 1

**B**unyavirales is a large order of segmented negative sense single-stranded RNA viruses (sNSV). It comprises 12 families and more than 500 viruses, amongst which human dangerous pathogens such as La Crosse (LACV, *Peribunyaviridae* family), Hantaan (HTNV, *Hantaviridae* family), Rift Valley Fever, Severe Fever with Thrombocytopenia Syndrome (RVFV and SFTSV, *Phenuiviridae* family), Lassa (LASV, *Arenaviridae* family), and Crimean Congo Haemorrhagic fever viruses (CCHFV, *Nairoviridae* family)[1,2]. There is currently no treatment or vaccine to counteract them. In this context, we focus our interest on LACV and more precisely on essential steps of its viral cycle: genome replication and transcription. These processes are catalyzed by the virally-encoded RNA-dependent RNA-polymerase (RdRp), also called L protein (LACV-L)[3]. This large 260 kDa multi-functional and monomeric enzyme performs replication of the viral genome (vRNA) into a complementary RNA (cRNA), which is then used as a template to generate nascent vRNA that have a size and composition identical to the parental vRNA. Replication initiation is performed de novo, in the absence of primer. LACV-L and other *Peribunyaviridae* are suspected to initiate their replication internally at position 4 of the RNA template to produce a primer that then realigns to the template end. This process, called "prime-and-realign", is made possible by a triplet nucleotide repetition at the 3′-vRNA template end (3′-UCAUCA…−5′ for LACV) and has been reported for several families in the *Bunyavirales* order, although with family-dependent specificities[4–6]. Concerning transcription, LACV-L catalyses its initiation through a cap-snatching mechanism, whereby LACV-L steals cellular capped mRNA, binds them through its cap-binding domain (CBD), cleaves them 9 to 17 nucleotides downstream the cap with its endonuclease domain (ENDO), before using them as a primer for transcription initiation[7,8]. The prime-and-realign mechanism has also been observed at this stage resulting in the insertion of a triplet (5′-AGU-3′) or multiple triplets (5′-AGUAGU…−3′) complementary to the 3′-template inserted between the cap-snatched mRNA primer and the viral RNA transcript[7,9,10].

Structural determination of these essential and complex sNSV polymerases has long been a challenge in the field. Following determination of the X-ray structure of the C-terminally truncated construct of LACV-L[11], structures of full-length LACV-L, LASV-L, Machupo-L (MACV-L), and SFTSV-L have been determined by cryo-EM[12–15]. They contain a conserved central core comprising the polymerase active site, an N-terminal ENDO and a flexible C-terminal region (CTER) that includes the CBD. They were solved in their apo (SFTSV-L) or pre-initiation form (LACV-L, Machupo-L, and LASV-L) with the 3′-vRNA promoters bound in a secondary site at the surface of the polymerases, away from the RNA synthesis active site. The 5′-vRNA promoter, present only in LACV-L structures, is bound as a hook in a distinct and specific site on the polymerase surface.

These structures were important advances in the field, but they do not shed light on the detailed mechanisms by which this key multi-functional molecular machine performs genome replication and transcription. In the present article, we uncover the conditions necessary for LACV-L in vitro activity and determine LACV-L structures stalled at specific stages of both replication and transcription revealing with unprecedented detail the critical elements triggering activation of these essential processes.

## Results

**LACV-L replication activity.** Visualization of LACV-L activity necessitated careful optimization of (i) the construct, (ii) the tag position, and (iii) the composition of the vRNA promoters. Using a mini-replicon system, LACV-L constructs with an N- or C-terminal tag were found to display low activity compared to a construct without a tag (Fig. 1a). Addition of a Strep-tag in an internal exposed loop called the California insertion (LACV-L$_{CItag}$) displayed 85% activity compared to the LACV-L without a tag and was thus chosen for subsequent purification and in vitro activity assays (Fig. 1a). To abolish unspecific RNA degradation in vitro, we introduced the H34K mutation known to prevent ENDO activity[16]. Despite the optimized construct, the homogeneously purified LACV-L$_{CItag\_H34K}$ (Supplementary Fig. 1a) incubated with wild-type (WT) 5′/3′-vRNA promoters, NTPs, and MgCl$_2$ at 30 °C for 4 h did not generate replication products (Fig. 1b, lane 1). This behavior can be explained by the high complementarity of the 5′/3′-vRNA promoters that tend to form stable double-stranded RNA, preventing their binding as single-stranded RNA at their specific binding site on LACV-L. This situation is specific to in vitro reconstitution as in vivo each promoter is kept in separate binding sites on the polymerase. To avoid 5′/3′-vRNA duplex formation, we analyzed the structure of the 5′-vRNA promoter and its interaction with the polymerase. We generated a 17-base-pair modified 5′ RNA (5′-1-17BPm), by mutating the nucleotides G2, U3, A9, and C10 of the 5′-end into C2, G3, C9, and G10, thereby preserving the hook structure and its interaction with LACV-L while significantly decreasing the 5′/3′-vRNA complementarity (Fig. 1c, pre-initiation vs initiation). Equivalent activity assays with the 5′-1-17BPm in place of the WT 5′-vRNA led to the formation of a 25-nucleotide replication product (Fig. 1b, lane 2, Supplementary Fig. 1b). Next-generation RNA sequencing (NGS) confirms the presence of expected 25-mer replication products (Supplementary Fig. 2a). It also identifies 25-mer products that contain a misincorporated terminal 5′ nucleotide and therefore start with GGU instead of AGU. Slower-migrating products are also visible on gel and NGS identifies them as being products extended in 5′ with GU and GGU sequences. These correspond to products in which realignment occurred twice. In addition, NGS identifies that some products are extended in 3′ by 3–6 nucleotides with the sequence CUU to CUUGGU. Visualization of replication activity with LACV-L$_{CItag\_H34K}$, 5′-1-17BPm and 3′-vRNA1-25 led us to optimize the assay revealing a maximal activity with 2 to 5 mM Mg$^{2+}$ while more abortive products and fewer 25-mer replication products are visualized in presence of Mn$^{2+}$ (Supplementary Fig. 1c). A time course indicates that the reaction is optimal at 4 h (Supplementary Fig. 1d). We then performed equivalent reaction assays omitting either the 5′- or the 3′-vRNA. These resulted in the absence of 25-mer product formation, clearly indicating the requirement of both promoter ends for replication activity (Fig. 1b, lanes 3, 4). The dependence of the product size on the template size was analysed, and reveals that, as expected, a 3′-vRNA1-30 template gives rise to products 5 nucleotides longer than a 3′-vRNA1-25 template (Fig. 1b, lanes 2, 5). Incubation of LACV-L$_{CItag\_H34K}$ with 5′-1-17BPm, 3′ vRNA1-25, ATP, UTP, and GTP generates a 9-mer product, consistently with the template sequence that requires CTP incorporation at position 10 (Fig. 1b, lanes 6, 7). Interestingly, CTP addition after 1 h restores complete product formation, indicating that the reaction performed with ATP, UTP, and GTP is stalled in a physiological, early-elongation state (Fig. 1b, lanes 8, 9). Higher molecular weight bands appear in addition to the expected products (Fig. 1b, lanes 2, 5, 8, 9) and likely correspond to the template/product hybridization upon assay arrest due to their high complementary, as reported in several sNSV in vitro activity assays[14,17].

**Overview of the actively replicating LACV-L structures.** Visualisation of replication activity prompted us to collect three large cryo-EM datasets which, coupled with advanced image

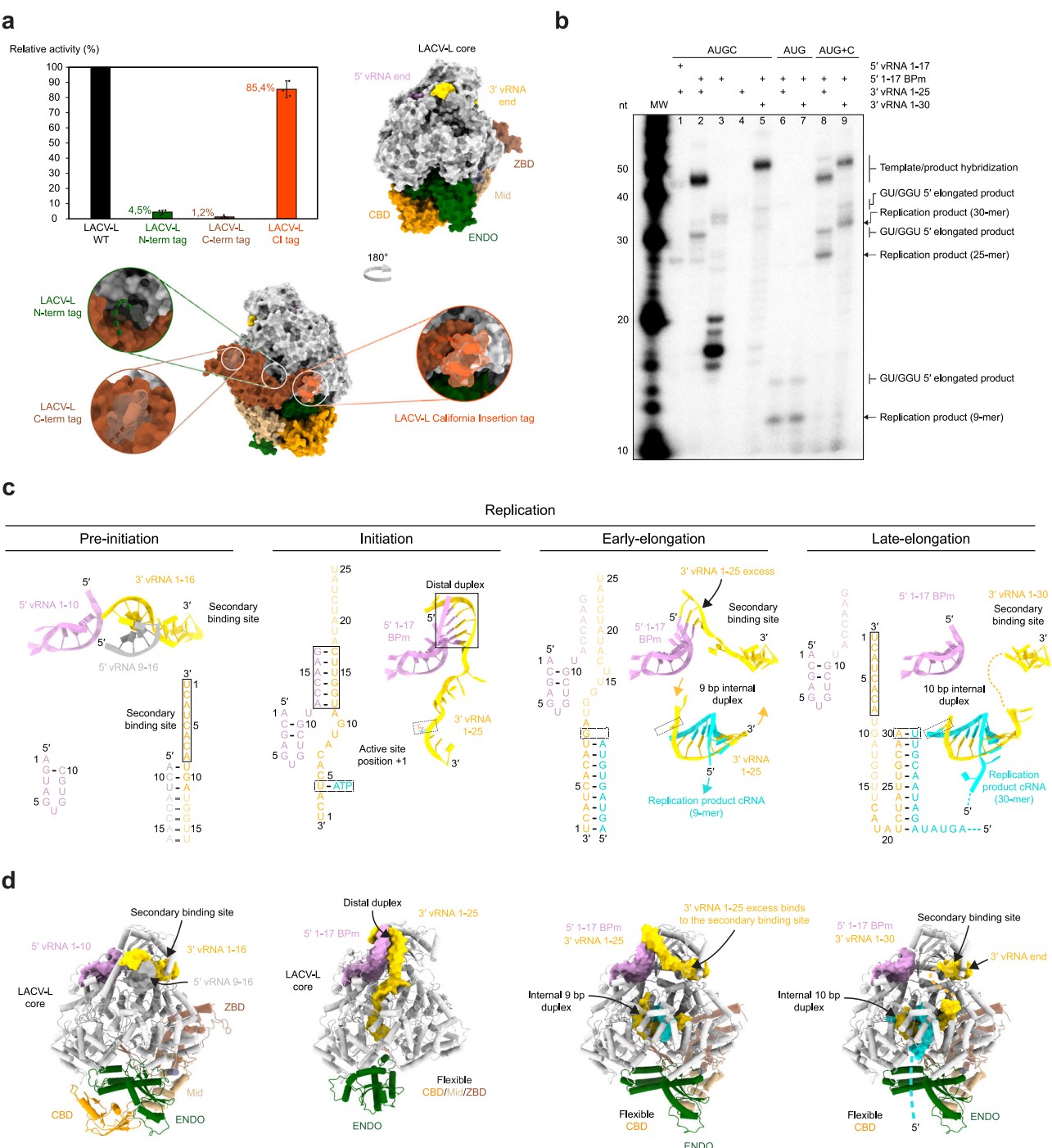

**Fig. 1 Overview of LACV-L activity and cryo-EM structures stalled in specific replication states. a** Histogram of LACV-L mini-replicon activity. The percentage of Renilla/Luciferase activity of the constructs containing the N-terminal hexa-histidine tag (N-term tag), the C-terminal hexa-histidine tag (C-term tag), and the California insertion tag (CI tag) are compared to LACV-L activity of the construct without tag (WT). Tag positions are shown on LACV-L (PDB: 6Z6G) as dotted lines. The polymerase core is colored in light gray with the CI in orange, the cap-binding domain (CBD) in gold, the endonuclease (ENDO) in green, the mid domain in beige and the zinc-binding domain (ZBD) in brown. $n = 3$ biologically independent experiments. Error bars are mean ± standard deviation. Source data are provided as a Source Data file. **b** In vitro replication activity of LACV-L$_{CItag\_H34K}$ using different combinations of 5′- and 3′-vRNAs. The assays are done in presence of either the 4 NTPs (AUGC), 3 NTPs (AUG) or 3 NTPs during 4 h, subsequently supplemented with CTP for 30 min (AUG+C). 5′-vRNA containing nucleotides 1 to 17 (5′ vRNA 1-17) or 5′-vRNA with base pair mutation (5′ 1-17BPm) is used. 3′-vRNA contain either nucleotides 1 to 25 (3′ vRNA 1-25) or 1 to 30 (3′ vRNA 1–30). Products have either the expected size or are extended in 5′ with GU or GGU (GU/GGU 5′ elongated products). Replication products and their respective lengths are displayed on the right side of the gel. The decade molecular weight marker (MW) is shown. Source data are provided as a Source Data file. This experiment was repeated independently 3 times with similar results. **c** Sequence and secondary structures of the RNA bound to LACV-L$_{CItag\_H34K}$ for each replication state. 5′-vRNAs, 3′-vRNAs and replication products are respectively colored in pink/gray, gold and cyan. Bases present in the sequences but not seen in the structures are shown in transparent. **d** Cartoon representation of each LACV-L$_{CItag\_H34K}$ replication structure. LACV-L domains are colored as in **a**. RNAs displayed as surfaces are colored as in **a**, **c**. The pre-initiation state corresponds to the PDB 6Z6G.

processing involving extensive 3D classifications, captured LACV-L structures in three key states of replication: initiation, early-elongation, and late-elongation at 2.8, 2.9, and 3.9 Å resolution, respectively (Fig. 1c, d, Supplementary Table 1, and Supplementary Movie 1). To stabilize the replication initiation state, LACV-L$_{CItag\_H34K}$ was incubated with the 5′-1-17BPm and the 3′-vRNA1-25 in presence of the initial nucleotides to be incorporated. An initiation-mimicking state, with ATP present at position +1 of the active site, was obtained when the LACV-L initiation complex was formed in presence of ATP and UTP (Fig. 1c, d and Supplementary Fig. 3). Although UTP was added in the mix, it is not observed in position -1, which appears to indicate that it is not incorporated at initiation. For the replication early-elongation state, LACV-L$_{CItag\_H34K}$ was incubated with the previously used RNAs supplemented with ATP, UTP, and GTP resulting in the formation of a 9-base pair elongating template-product duplex in the polymerase internal cavity (Fig. 1c, d and Supplementary Fig. 4a). For the late-elongation state, a 5′-1-17BPm, 3′-vRNA-1-30 and four nucleotides were used. In this state, the 3′-vRNA is entirely replicated, a 10-base pair template-product duplex is visible in the active site and the 3′-vRNA end, following its exit from the active site as a single-stranded RNA, binds in the polymerase 3′-end secondary binding site (Fig. 1c, d and Supplementary Fig. 4b).

Comparison of the three structures reveals major conformational changes of LACV-L during initiation compared to its stable conformation found in the pre-initiation and the early-/late-elongation states (Fig. 1d). A major movement of the ENDO is visible at initiation, coordinated with the movement of the entire CTER, including the CBD, that becomes flexible at this stage. Whereas the ENDO, mid and zinc-binding domain (ZBD) gain back their pre-initiation position at elongation, the CBD remains flexible.

**The replication initiation state**. The principal novelty in the obtained structures concerns the initiation conformation that provides molecular insights into the mechanics underlying this critical step. The distal duplex, formed between nucleotides 12 to 17 of the 3′/5′-vRNA ends, plays a critical role in triggering the template positioning in the RNA tunnel entrance (Fig. 2a and Supplementary Fig. 5a). The distal duplex is held in place by the 5′-vRNA hook (nucleotides 1-10), the vRNA Binding Lobe domain (vRBL), the clamp, the arch, and the α-ribbon. The RNA entrance tunnel is surrounded by the vRBL, the fingers, the thumb ring, and the bridge domains. Nucleotides G9 to A11 of the template are stabilized in the tunnel by multiple hydrophobic and polar interactions while nucleotides A6, C7, and A8 are less coordinated and adopt alternative conformations (Supplementary Fig. 5a). Their visualized mobility might provide the template with the necessary flexibility for its realignment after priming (Fig. 2b). The 3′ terminal nucleotides U1 to C5 display clearer density correlated with their higher degree of stabilization by the finger domain and in particular the canonical polymerase motif F (Fig. 2c and Supplementary Fig. 5a). The 3′ terminus itself interacts by an unexpected but crucial element: the loop 982-995, which is localized between the fingers and the palm domain, and which, unexpectedly, plays the role of a replication priming loop (Fig. 2c). Its dual role in priming and realignment (see below) leads us to name this element the prime-and-realign loop (PR loop).

At initiation, the ENDO conformational change brings its residues 172–184 (α-helix 7) proximally to the PR loop (Fig. 2c). The ENDO residues E177 and K181 would clash with the PR loop in the absence of its rearrangement, suggesting that the ENDO movement is linked with the PR loop extension. In this extended

state, the PR loop residue M989 interacts with the 3′ terminal nucleotide U1 in position −3 of the active site while the residues I990 and S991 stabilize the nucleotide C2 in position −2 of the active site (Fig. 2c). As the result, the template nucleotides A3 and U4 are localized in position −1 and +1 of the active site (Figs. 1c and 2c). The map displays a clear density for an ATP in position +1 that makes hydrogen bonds with U4 and stacks with A3 (Supplementary Fig. 5a). Its ribose interacts with W1064 (from motif A) and Q1145 (from motif B), its phosphates interact with D1060, M1061, K1063 (from motif A), D1187 (from motif C) and two magnesium ions, themselves located in canonical positions for catalytic magnesium A and B (Fig. 2c).

The identification of the unexpected PR loop led us to analyze further its role in replication by engineering single-alanine substitutions of its tip residues M989, I990, and S991 (Fig. 2e). M989A diminishes the formation of 25-mer product by 95% and of GU/GGU 5′ extended replication products by 87%, confirming the importance of this residue in precise template positioning at initiation (Fig. 2e, lane 2). I990A mutation multiplies by 21.6 and 7.93 the formation of respectively 25-mer and GU/GGU 5′ extended replication products (Fig. 2e, lane 3 vs lane 1). LACV-L$_{CItag\_H34K\_S991A}$ produces 13.8 times more GU/GGU 5′ extended replication products than LACV-L$_{CItag\_H34K}$. Altogether, these mutations clearly confirm the importance of the PR loop tip in the replication mechanism with a defined role of M989 at initiation and an unexpected role of the mutations I990A and S991A in the formation of 25-mer and/or GU/GGU 5′ extended replication products.

**Switch to elongation and related conformational changes**. Progression towards elongation implies important remodeling of LACV-L domains. Retraction of the PR loop, which is coupled with ENDO repositioning (Figs. 2d and 3a), provides space for the initial formation of the template-product duplex in the active site cavity. The absence of CTP in the mix results in LACV-L stalling with template G10 in position +1 of the active site in a post-translocation pre-incorporation state (Fig. 2d and Supplementary Fig. 5b). Whereas the replication initiation was performed internally, realignment must have occurred, as the replication elongation displays an entire product with the 5′-cRNA end that is complementary to the 3′-vRNA nucleotide 1. To accommodate the internal 9-base pair duplex, switch from initiation to elongation state results in the opening of the lid, the thumb, and the thumb-ring coupled with the extension of the bridge loop 1423–1441 by unwinding of the helix 1435–1438 (Fig. 3b). Together these movements switch the core from a closed to an open state, resulting in the extrusion of the previously defined "priming loop" that induces the opening of the template exit tunnel (Fig. 3c). As the "priming loop" remains away from the active site during replication and does not appear to be involved in priming in the visualized states, it is renamed "template exit plug" (Fig. 3b, c). Interestingly, reverse movement from an open to a closed core is observed between pre-initiation and initiation (Fig. 3b, c) suggesting that the core closure is a rather transient state acquired at initiation only, while the open core is a more stable long-standing state.

In addition to the core opening/closing between the replication states, global movements of the vRBL, clamp, arch, and α-ribbon are visualized (Fig. 3b, c). At pre-initiation, the α-ribbon is ordered and the vRBL is proximal to the core, thereby forming a cleft called the 3′-vRNA secondary binding site to which the 3′-vRNA end binds specifically. At initiation, the acquired α-ribbon flexibility and displacement of the vRBL away from the core disrupt the 3′-vRNA secondary binding site. The 3′-vRNA changes its position to orient itself

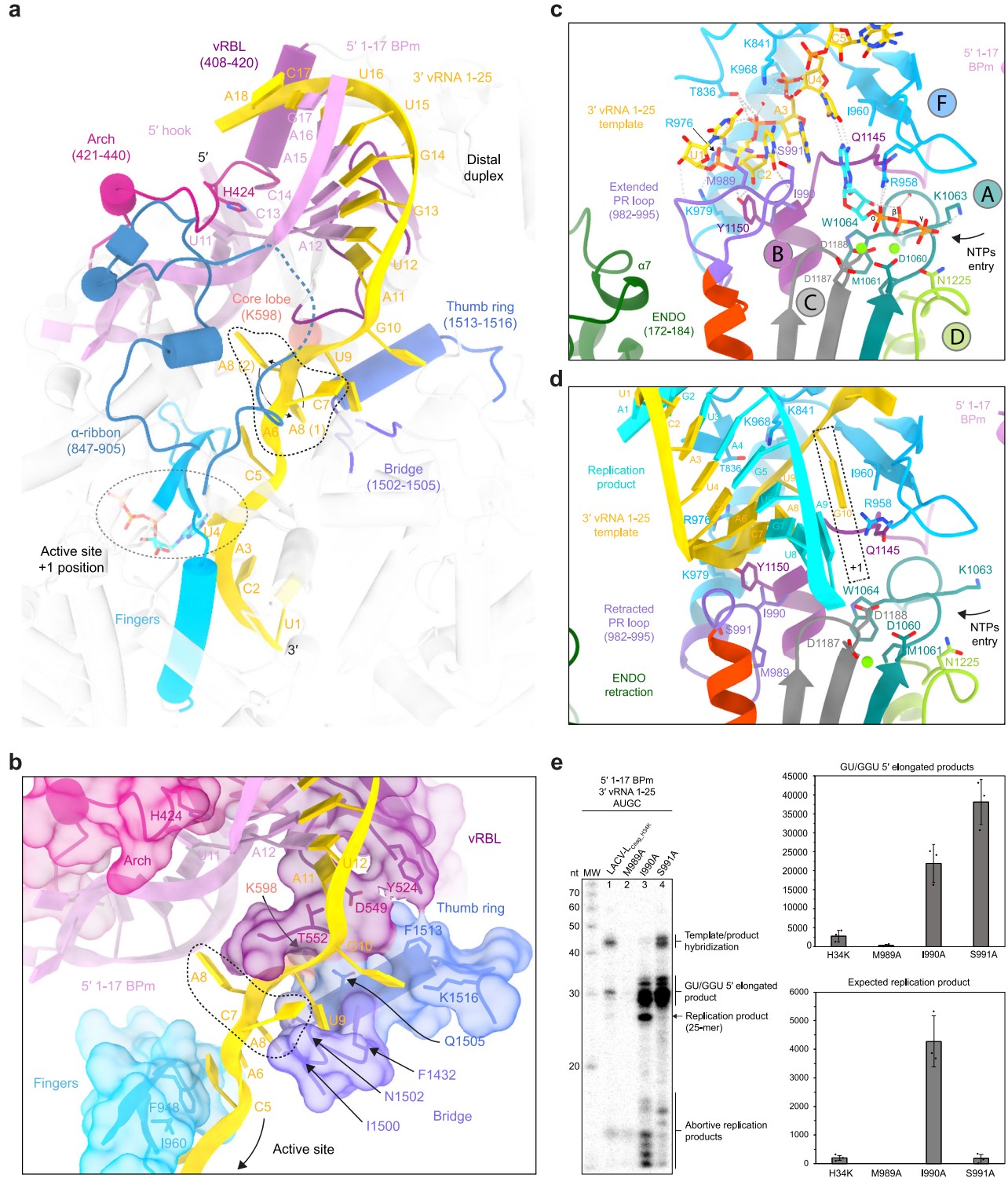

into the template entry tunnel necessary for initiation. At late-elongation stage, the exit of the 3′ template from the template exit channel is coordinated with a repositioning of the vRBL and the clamp close to the core resulting in the formation of the 3′ secondary binding site as in pre-initiation. The replicated 3′-vRNA end exits as a single-stranded RNA from the active site cavity, binds to a positively charged groove (Supplementary Figs. 5c and 6), before reaching the 3′ secondary binding site, ready for the next replication cycle.

**LACV-L transcription activity.** Transcription activity was analysed using LACV-L$_{CItag\_H34K}$ incubated with 14-mer capped RNA primers, the 3′-vRNA1-25/1-30 and the 5′-1-17BPm. The chosen capped RNA primers have a two- (cap14AG) or three-nucleotide (cap14AGU) complementarity with the 3′-vRNA end (5′-…ACUACU-3′) to facilitate priming. Time, divalent ion identity and divalent ion concentration were tested and indicate that the transcription reaction is optimal after 30 min in the presence of 2 mM MgCl$_2$ (Supplementary Fig. 7a, b). Using these conditions,

**Fig. 2 Interactions between LACV-LCltag_H34K and 5′/3′-vRNA during replication. a** Interaction between LACV-L$_{Cltag\_H34K}$ and the distal duplex formed by 5′-1-17BPm and 3′-vRNA1-25. Interacting domains, namely the arch, the vRNA-binding lobe (vRBL), the α-ribbon, the fingers, the bridge and the thumb ring domains are respectively colored in magenta, violet red, steel blue, cyan, purple and blue. The RNAs are colored as in Fig. 1c, d. ATP in position +1 of the active site is surrounded by a dotted line. **b** Zoom on the interaction between LACV-L$_{Cltag\_H34K}$ and the 3′-vRNA1-25. Surface of interacting domains are displayed and colored according to **a**. Residues that interact with the 3′-vRNA template are labeled. Flexible nucleotides A6/C7/A8 of the 3′-vRNA1-25 are surrounded by a dotted line. The two visible conformations of A8 are shown. **c** Active site of LACV-L$_{Cltag\_H34K}$ in replication initiation state. RdRp motifs A, B, C, D, and F are respectively colored in dark turquoise, purple, gray, light green, and blue. The extended prime-and-realign loop (PR loop) and ENDO are colored in orchid and green. Residues that interact with the 3′-vRNA1-25 are displayed. The two magnesium ions are colored in light green. **d** Active site of LACV-L$_{Cltag\_H34K}$ in replication elongation state. LACV-L domains, RdRp motifs, and vRNAs are colored as in **c**. The newly synthesized product forming a 9 base-pair duplex with the 3′-vRNA1-25 template is colored in cyan. The +1 active site position is surrounded by a dotted line. **e** In vitro replication activity of LACV-L$_{Cltag\_H34K}$, LACV-L$_{Cltag\_H34K\_M989A}$ (M989A), LACV-L$_{Cltag\_H34K\_I990A}$ (I990A), LACV-L$_{Cltag\_H34K\_S991A}$ (S991A) with 5′-1-17BPm, 3′-vRNA1-25 and 4 NTPs. Replication products and their respective lengths are displayed on the right side of the gel. Histogram of product formation. Intensities are mean ± standard deviation. $n = 3$ biologically independent reactions. Quantitative comparison come from two different gels that have been processed in parallel. Source data are provided as a Source Data file. P-value for two-tailed unpaired t-test between of LACV-L$_{Cltag\_H34K}$ and each LACV-L PR loop mutant are shown. $P < 0.05$ is considered significant. Intensity ratios (LACV-L PR loop mutant/LACV-L$_{Cltag\_H34K}$) are displayed.

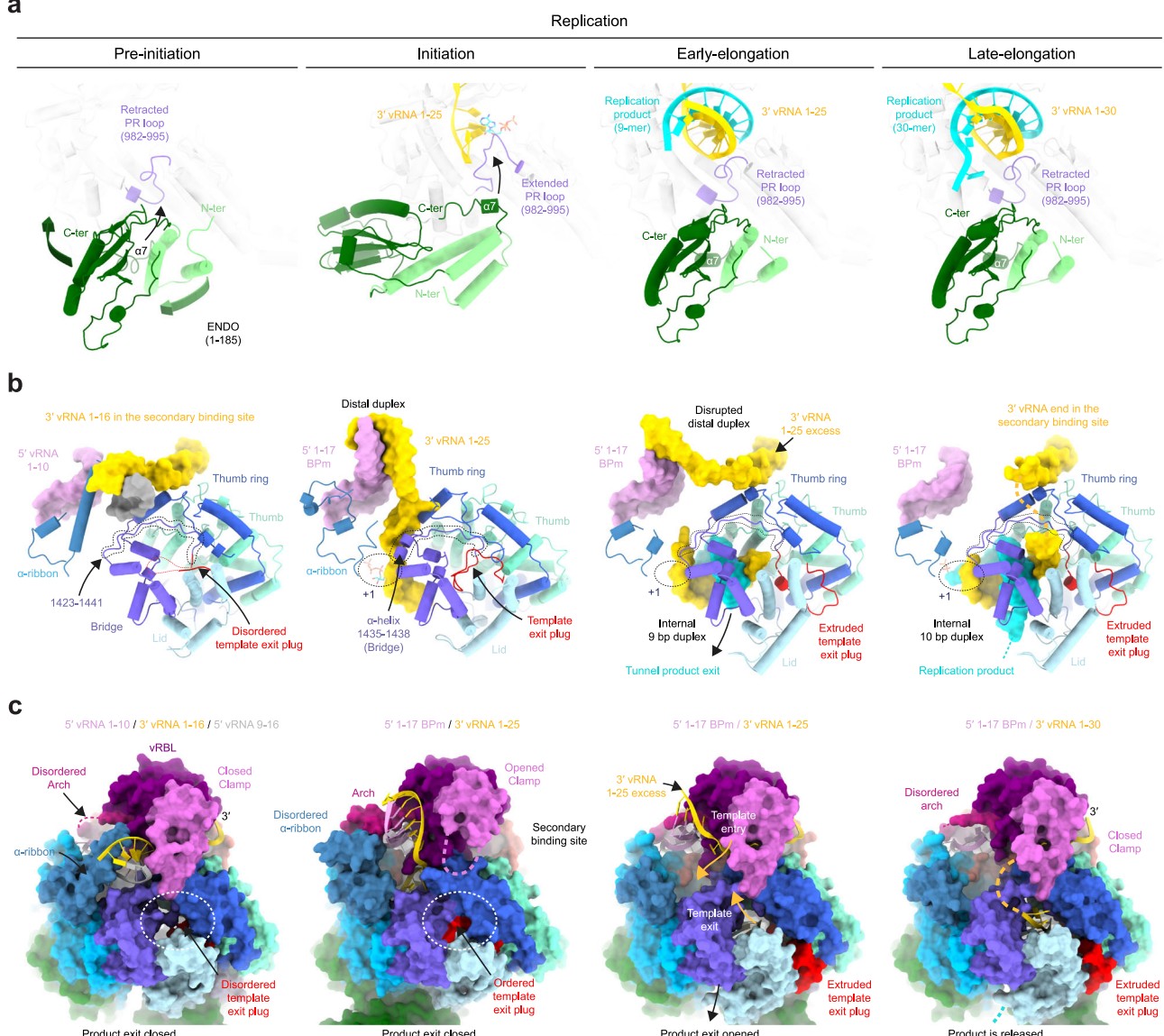

**Fig. 3 Replication-induced conformational changes of LACV-LCltag_H34K. a** Endonuclease (ENDO) and prime-and-realign loop (PR loop) movements between the replication states. ENDO N-terminus (N-ter) and C-terminus (C-ter) are indicated. Arrows indicate movements of the ENDO, the α7 helix, and the PR loop. **b** LACV-L domain movements and RNA position variations between the replication states. For clarity, only domains that undergo conformational changes are displayed. The template exit plug, colored in red, is indicated with a dotted line if flexible. RNAs are shown as surfaces and colored as in Fig. 1c, d. Bridge residues 1423–1441 are surrounded by a dotted line. **c** Template exit plug, vRNA binding lobe (vRBL), clamp, and α-ribbon conformational changes between the replication states. LACV-L surface is displayed and each domain is colored as in Figs. 2a, b and 3b. The template exit tunnel is surrounded by a dotted line. Template entry/exit tunnels are indicated in the early-elongation state with gold arrows.

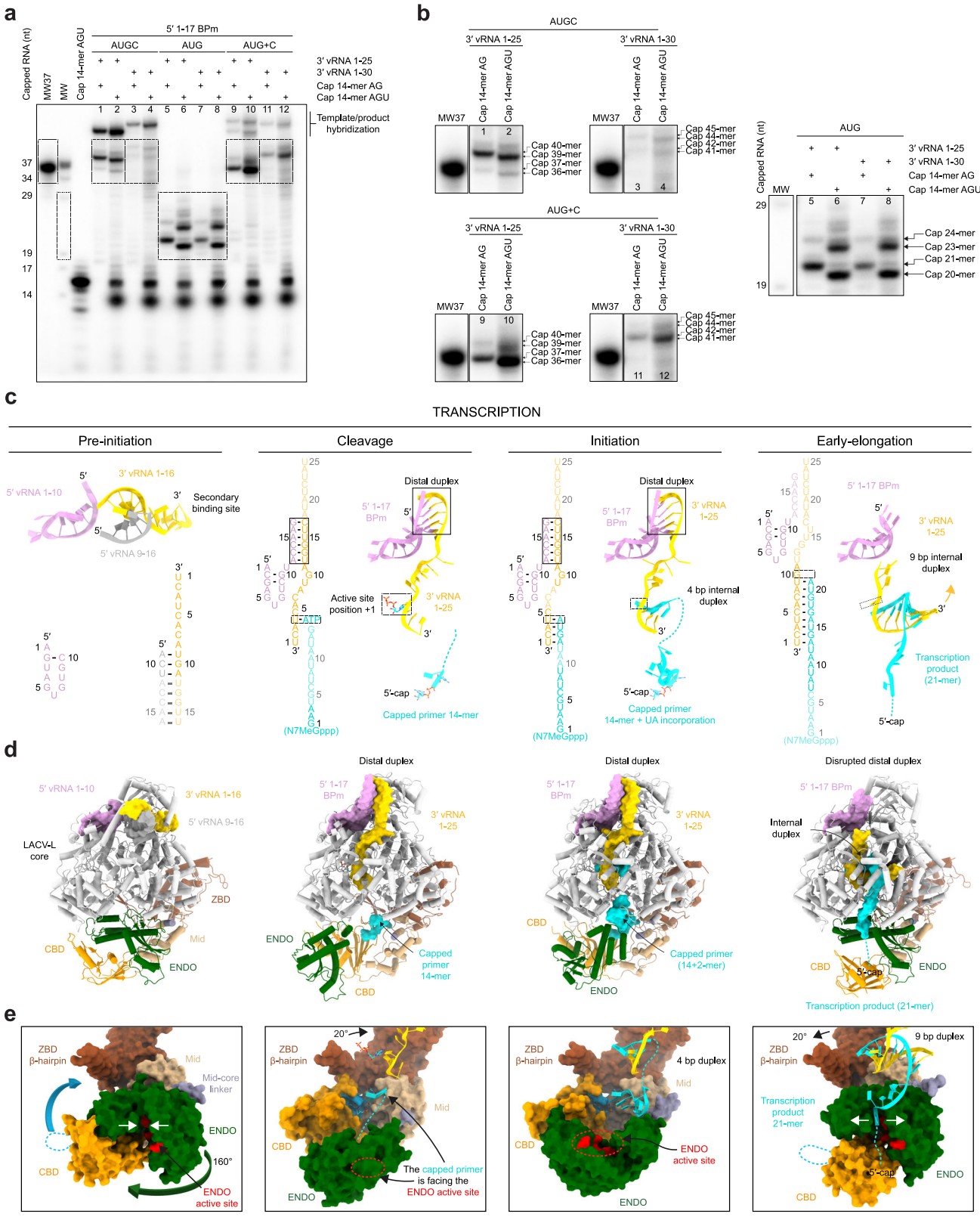

the reactions with cap14AG/cap14AGU lead to the formation of two main transcription products, a 37-/36-mer and a 40-/39-mer (Fig. 4a, b, lanes 1, 2). The minority 37-/36-mer products correspond to the capped RNA size elongated by 23/22 nucleotides, considering the hybridization of the last 2 or 3 template nucleotides with the cap14AG/capAGU. Their migration is consistent with a

capped molecular weight marker of the same size and composition, and they are identified in NGS (Fig. 4a, b). The majority 40-/39-mer products correspond to primed and subsequently realigned transcripts, which result in the addition of an AGU nucleotide triplet at the end of the capped primer before elongation, as detected by NGS analysis (Supplementary Fig. 2b). NGS also reveals the presence of

**Fig. 4 Overview of LACV-LCItag_H34K activity and cryo-EM structures in transcription. a** In vitro transcription activity of LACV-L$_{CItag\_H34K}$ using different combinations of 5′/3′-vRNAs, NTPs and capped primers (capped 14-mer finishing by "AG" (cap14AG) and in vitro produced capped 14-mer finishing by "AGU" (IVT cap14AGU)). The MW37 lane corresponds to a capped 37-mer identical in sequence to the theoretical transcription product of LACV-L$_{CItag\_H34K}$ with the cap14AG and the 3′-vRNA1-25 without prime-and-realign. The lane indicated as MW corresponds to a RNASe T1 cleavage of MW37. Dotted squares indicate transcription products of interest that are reported in **b**. Source data are provided as a Source Data file. This experiment was repeated independently 3 times with similar results. **b** Zoom on LACV-L$_{CItag\_H34K}$ transcription products. Lane numbers are referring to **a**. Transcription products length are indicated on the right side of the cropped gels with the corresponding molecular weight ladder. Source data are provided as a Source Data file. **c** Sequence and secondary structures of 5′/3′-vRNAs, capped RNA primer and transcription products. 5′-vRNAs, 3′-vRNAs and capped primer/product are respectively colored in pink/gray, gold and cyan. Bases present in the sequences but not seen in the structures are shown in transparent. **d** Cartoon representation of each LACV-L$_{CItag\_H34K}$ transcription structure. Note that LACV-L in "capped primer active site entry" state has the same conformation as in "initiation" state. LACV-L domains are colored as in Fig. 1. RNAs are displayed as surfaces and colored as in **c**. **e** ENDO and CBD movements during transcription. The ENDO, CBD, Mid/ZBD, RNAs are displayed as surfaces and colored as in **d**. The ENDO active site and CBD cap-binding site are colored in red and blue. The closing/opening of product exit tunnel is highlighted by white arrows.

products that have been realigned up to three times. This is in accordance with the sequencing of capped RNA produced by *Peribunyaviridae* polymerase in vivo[7,10,18]. As expected, the size of the transcribed RNA depends on the length of the template: replacing the 3′-vRNA1-25 by a 3′-vRNA1-30 gives rise to products that are 5 nucleotides longer (Fig. 4a, b, lanes 3, 4). Usage of an NTP subset containing ATP, UTP, and GTP, to obtain an early-stalled transcription complex, shows formation of the expected 21/24-nucleotide products with cap14AG and 20/23 nucleotide products with cap14AGU (Fig. 4a, b, lanes 5–8, and Supplementary Fig. 7c). Interestingly, whereas the realignment was almost complete in presence of the 4 NTP, it is only partial in presence of the 3-NTP subset. Subsequent incubation with a CTP restores the formation of full-length capped RNA product, but with a majority of non-realigned capped RNA products (Fig. 4a, b, lanes 9−12).

**Overview of the transcriptionally active LACV-L structures**. To visualize the path taken by the RNA during transcription initiation, a large cryo-EM dataset was collected on a transcription reaction mix containing LACV-L$_{CItag−H34K}$, the cap14AG, the 3′-vRNA1-25, the 5′-1-17BPm, UTP, ATP, and MgCl₂. Advanced image processing enabled us to separate and reconstruct several active snapshots (Supplementary Figs. 3 and 8, Supplementary Table 2, and Supplementary Movie 2). A conformation captured at 3.9 Å resolution shows the capped RNA bound to the CBD, 45 Å away from the ENDO, a distance compatible with capped RNA primer cleavage after 9–17 nucleotides (Fig. 4c, d). We called this snapshot the "putative endonuclease cleavage conformation", although this remains speculative. Only the first two nucleotides of the capped RNA are visible, likely due to the low affinity of the ENDO for RNA, which is related to the diversity of sequences to be cut (Supplementary Fig. 9a). The 5′/3′ promoters are properly positioned for transcription initiation (Supplementary Fig. 9a). The second conformation, obtained at 3.1 Å resolution, shows the cap and the first two nucleotides of the capped RNA protruding towards the active site (Supplementary Fig. 9b). It likely corresponds to the conformation just prior transcription initiation and was therefore called "capped primer active site entry". A "transcription initiation conformation" structure at 3.6 Å resolution is also trapped for a small-particle subset (Fig. 4c, d). The capped RNA is visible in the CBD and in the active site with flexible nucleotides in between. It shows incorporation of UTP and ATP in the capped RNA product (Supplementary Fig. 9c). Finally, another cryo-EM data collection and image processing led to the determination of a transcription "early-elongation" conformation structure at 3.3 Å resolution. It was obtained by incubating the transcription complex for 30 min at 30 °C in presence of ATP, UTP, GTP, and MgCl₂. It shows an elongated capped RNA that forms a 9-base pair template-product

duplex in the active site cavity (Fig. 4c, d and Supplementary Fig. 9d).

**Major functionally relevant conformational changes of LACV-L are visualized between the transcription states**. The switch from the pre-initiation to the putative endonuclease cleavage conformation is coupled with a large ENDO movement resulting in a 160° rotation that liberates the previously blocked capped RNA tunnel entrance (Fig. 4e). The acquired ENDO position is reminiscent to the orientation visualized in the LACV-L ΔCTER X-ray and cryo-EM structures[11]. In addition to the ENDO movement, the entire CTER rotates by 20° (Fig. 5a), bringing the ZBD and the mid domain towards the core, where they interact with the thumb ring (Fig. 5b). This conformation of the CTER brings the CBD binding site close to the capped RNA entrance tunnel, contrasting with its exposed position taken at pre-initiation (Fig. 4e). The CBD, and in particular the loop 1850–1859 that is involved in cap binding and was disordered at pre-initiation, is stabilized by several domains and becomes visible, explaining the proper capped RNA binding site formation (Fig. 5b). The CBD interacts with the palm domain, the mid, the ZBD β-hairpin and the core lobe.

Switch from the endonuclease cleavage conformation to the transcription initiation conformation involves a 175° rotation of the ENDO (Fig. 4d, e). Interestingly, ENDO charge analysis suggests that, whereas the negatively charged capped RNA might be attracted by the ENDO positively charged catalytic site in the endonuclease cleavage conformation, it would be repelled by the negatively charged ENDO surface in the transcription initiation conformation, thereby facilitating capped RNA entry towards the polymerase active site (Supplementary Fig. 10). In addition, to the large ENDO and CTER movements, switch from pre-initiation to endonuclease cleavage conformation/transcription initiation is coupled with core closure, in an analogous manner to what is observed at replication initiation.

Nearly all the conformational changes that resulted in transcription initiation are reversed at elongation, bringing back the core in an open conformation, the ENDO and the CTER in the pre-initiation conformation (Fig. 4d, e). The only exception concerns the CBD that acquires an orientation differing of 30° compared to pre-initiation.

**Zoom on contacts between LACV-L and RNA during transcription**. The cleavage conformation reveals the peculiar binding of the cap that is stacked between W1847 and R1854 of the CBD, with Q1851 ensuring guanine specificity (Fig. 6a). The cap triphosphate moiety makes hydrogen bond interactions with K2011, K2012, R1854, and W1850. The two next nucleotides interact through their bases with Y714 of the core lobe domain, H2014, F2015, and K2017 of the mid domain and Y1716 of the thumb

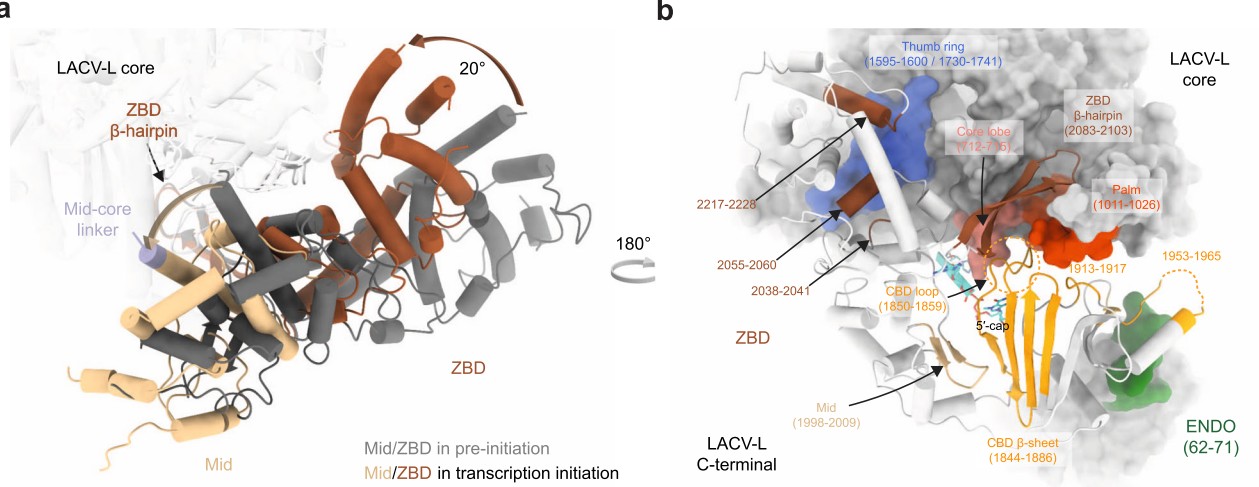

**Fig. 5 LACV-LCItag_H34K conformational changes induced by transcription initiation. a** Mid and ZBD movements from pre-initiation to transcription initiation states. LACV-L core is transparent. Mid and ZBD are colored in gray in the pre-initiation state, in beige and brown in the transcription initiation state and their rotation highlighted with an arrow. **b** CTER interaction with the polymerase core and the ENDO during transcription initiation. Both LACV-L core and ENDO are displayed as gray surface. The CTER domains (mid, CBD, ZBD) are displayed as cartoon. All interacting regions are displayed, colored and numbered. The CBD loop close to the CBD-binding site is surrounded by a dotted line.

ring domain, while the phosphate is stabilized by K2017 (Fig. 6a). The cleavage conformation also reveals that the PR loop tip stabilizes the template 3′ terminus prior to capped entry into the active site (Fig. 6a).

Following template binding and capped RNA cleavage, the capped RNA is directed towards the active site (Fig. 6b). The cap and the first 2 nucleotides of the primer remain bound in a similar fashion as in the cleavage conformation. The next nucleotides more sparsely bind through their phosphates to charged residues of the finger domain (R820, R824, K983), the thumb-ring (R1614), and the lid (H1620, H1621, D1641) (Fig. 6b). Because these residues bind to the phosphate backbone, it is likely that binding is sequence independent and can probably adapt to various sizes of snatched capped RNA primer. The RNA density is more defined towards the active site, indicating tighter binding (Supplementary Fig. 9c). Capped RNA bases 13A and 14G that are complementary to 1U, 2A of the 3′-vRNA end form a duplex in position −3 and −2 of the active site (Fig. 6b). The UTP and ATP nucleotides added to the reactions were incorporated into the nascent mRNA and are positioned in position −1 and +1 of the active site in front of 3′-vRNA A3 and U4.

Following the transcription initiation state visualized in the structure, a realignment must occur, as revealed by in vitro transcription products that are three nucleotides longer than if the capped RNA was simply elongated (Fig. 4a, b). We thus investigated the structure at initiation to identify which part of the polymerase might be implicated in realignment. Interestingly, the PR loop is in a retracted position and is proximal to the template position −4. PR loop-induced template realignment from position −4 to position −1 would perfectly reposition the template for subsequent elongation (Fig. 6b). We thus investigated if mutation of the PR loop tip might affect realignment. While transcription assays performed with LACV-L$_{CItag\_H34K\_M989A}$ and LACV-L$_{CItag\_H34K\_S991A}$ are equivalent to LACV-L$_{CItag\_H34K}$, LACV-L$_{CItag\_H34K\_I990A}$ displays a different transcription profile (Fig. 6c). NGS analysis reveals that LACV-L$_{CItag\_H34K\_I990A}$ generates, in addition to the expected 37- and 40-mer products, a significant number of aberrant products. These latter mainly correspond to the capped primer elongated by repetitive of AGU triplets that are likely added by realignment. Altogether these

results confirm the role of the PR loop, and in particular of the residue I990, in realignment during transcription activity.

Transcription then proceeds to elongation. While the template behaves like in replication, the capped RNA product exits through a tunnel that is different from the one used for its entry, surrounded by the bridge (K1474), the fingers (R820, N823, R824, R829), the lid (S1622, Y1696), and the thumb ring (H1704) (Fig. 6d). The capped RNA reaches the ENDO with which it interacts through residues (R33, K94, F162, H184). The 5′-cap detaches itself from the CBD, permitting the exit of capped RNA through a cleft between the ENDO and the CBD.

## Discussion

Replication initiation using a prime-and-realign mechanism has been observed in activity assays for decades for several bunyaviruses[4–6] and is also used by Influenza for vRNA synthesis from cRNA[19]. However, it had not yet been structurally characterized in any of these viruses due to the flexibility of the 3′-cRNA end[20]. Here, we identify the elements stabilizing the 3′-vRNA up to the active site: (i) the distal duplex, corroborating its importance in replication assays[21–23], (ii) the newly identified PR loop.

The PR loop is present in its retracted position in the other structurally determined sNSV polymerases (Supplementary Fig. 11a). It remains to be shown whether its extension at initiation also occurs in other sNSV. In terms of sequence, it is not very conserved within Bunyaviruses, except for Hantaviruses L proteins which contain at the loop tip a tyrosine, isoleucine, and serine in place of M989, I990, S991 in LACV-L (Supplementary Fig. 11). As M989 is likely to stabilize the incoming base in position -1, a tyrosine may play an equivalent role in hantaviruses. The exact mechanism of initiation may however slightly differ between LACV-L and HTNV-L. Indeed, the 5′ end of HTNV RNA products have been shown to be monophosphate[4,24] whereas we show here that LACV-L products have triphosphate ends (Supplementary Fig. 1e). The model proposed for HTNV-L, that would need to be validated experimentally, is that an extra nucleotide in 5′ is added in the prime-and-realign process, that would then be cleaved, leaving a monophosphate at the 5′ end. Hantavirus PR loop might thus play an additional role at

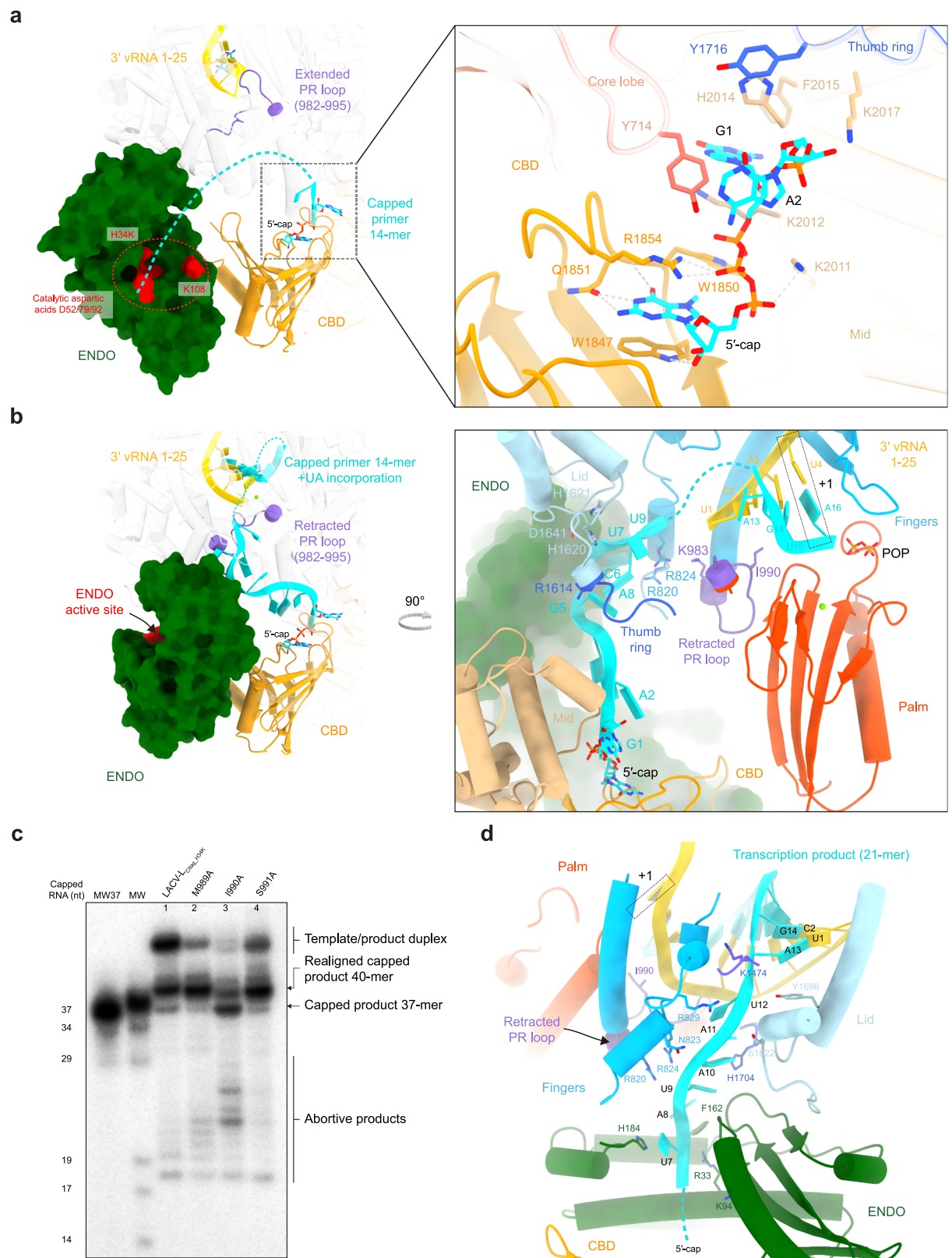

replication initiation. When comparing with Influenza ortho-myxovirus, the PR loop sequence is not conserved, but PB1 residue V273, which corresponds to the PR loop tip, is essential for prime-and-realign both in vitro and in vivo[25] (Supplementary Fig. 11). In Influenza, the PR loop may somehow complement the role of the priming loop in replication initiation[23,26,27].

Altogether, these elements strongly suggest the importance of the PR loop for prime-and realign initiation, with family-related specificities.

In LACV-L, the PR loop extension at replication initiation is coupled with a conformational change of the ENDO (Fig. 3a and Supplementary Fig. 12a). In the case of influenza polymerase,

**Fig. 6 Zoom on essential RNA-LACV-L contacts during transcription. a** 5′-cap interaction with LACV-L$_{CItag\_H34K}$ in the transcription cleavage conformation. The endonuclease (ENDO) active site residues are colored in red. The capped primer is colored in cyan and the dotted line stands for the potential cleavage path. The position of the prime-and-realign loop (PR loop) and the cap-binding domain (CBD) are shown respectively in purple and orange. On the right panel, residues implicated in the 5′-capped stabilization are displayed. **b** Capped primer path from the CBD active site to the RdRp active site during transcription initiation. On the right panel, residues that are proximal to the capped primer are indicated. POP corresponds to the pyrophosphate product. **c** In vitro transcription activity of LACV-L$_{CItag\_H34K}$, LACV-L$_{CItag\_H34K\_M989A}$ (M989A), LACV-L$_{CItag\_H34K-I990A}$ (I990A), LACV-L$_{CItag\_H34K\_S991A}$ (S991A) with 5′-1-17BPm, cap14AG, 3′-vRNA1-25 and four NTPs. Transcription products are labeled. MW37 and MW lanes are equivalent to Fig. 4a description. Source data are provided as a Source Data file. This experiment was repeated independently 2 times with similar results. **d** Interaction between the capped RNA product and LACV-L$_{CItag\_H34K}$ in transcription elongation state. Residues stabilizing it are indicated. Flexible nucleotides are represented as a dotted line.

although template realignment occurs following internal initiation during cRNA to vRNA replication, the situation seems to be quite different. It is thought that two distinct influenza polymerase dimers, between an apo-polymerase and the replicating polymerase, are involved in this process, whereas for LACV-L, no dimers have yet been observed. One influenza dimer has been suggested to facilitate template realignment by modifying the position of influenza priming loop[20] (Supplementary Fig. 12b, c). The second replicase dimer depends essentially on the host factor ANP32, which mediates formation of dimer between a 'packaging' polymerase and the replicating polymerase[28]. Influenza polymerases in both these dimers are in the 'replicase' rather than 'transcriptase' conformation, which involves a re-positioning of the ENDO and re-organization of the PB2-CTER that brings the PB2-cap and the residues 490–493 of the Cap-627 linker close to the putative PR loop (Supplementary Fig. 12d). It remains to be seen if the residues 490–493 of the Cap-627 linker and the CBD may, when acquiring this position, trigger a rearrangement of the PR loop that could complement the role of the priming loop in the prime-and-realign mechanism.

Proper transcription initiation, on top of the 3′ template proper positioning, depends on capped RNA binding, cleavage, and positioning in the active site. In LACV-L, major conformational changes of the ENDO and the CTER are involved in this process. These movements are more pronounced than for influenza polymerase, in which the ENDO remains in the same position between the cleavage and the transcription initiation conformations, the only major movement being done by the PB2-cap that rotates of 70° to bring the capped RNA first to the ENDO active site and then to the polymerase active site[23,29]. The CBD and ENDO of LACV and influenza polymerases are in different positions compared to the core precluding their exact comparison but the distance separating the cap and the ENDO cleavage site are of 45 Å for LACV and 49/52 Å for Influenza, in accordance with the similar sizes of capped primers (Supplementary Fig. 13). Although the CBD positions differ, the capped RNA tunnel entrance is conserved, together with core closing upon initiation. However, the details of transcription initiation differ in the two polymerase due to the exact positioning of the 3′-template in the active site at initiation, with the first nucleotide of the vRNA template being in position -2 and -3 for Influenza and LACV respectively[26] (Supplementary Fig. 13).

Combination of in vitro activity assays and structural snapshots of LACV-L in different active conformations lead us to propose a structural model for the prime-and-realign mechanism (Fig. 7a). Concerning replication, we hypothesize that proper closing of the active site motifs coupled with the visualized flexibility of the nucleotides 6, 7, 8 of the 3′ template enable a transient translocation of both the template and ATP of 1 nucleotide, thereby positioning the ATP in position −1 of the active site, where it would interact with the residue M989 of the PR loop tip (Fig. 7b). This would enable positioning of a GTP in position 1 of the active site and formation of a nascent pppApG

product. Normal elongation would generate a pppApGpU product concomitantly with translocation of the template with its terminus located in position −5 of the active site. At this stage, the template would be in extreme tension due to (i) the presence of the distal duplex linked to the 5′-hook at the tunnel entrance, (ii) the maximum stretching of template nucleotides 6-8 and (iii) the template positioning constrains by the PR loop. These would act as a spring and break the hydrogen bonds between pppApGpU and the template nucleotide 4 to 6, repositioning the primer in front of template nucleotide 1 to 3, completing the realignment. Further elongation results in proper product formation, exactly complementary to the template.

Concerning transcription, it is expected that the template and capped RNA positioning are more permissive than in replication, to adapt to the capped RNA length and composition (Fig. 7c). Capped RNA terminating by AG would position in front of the 3′ template end in position −2 and −1 of the active site, ready for incorporation of a U in position 1. This would require the retraction by one nucleotide of the template, made possible by the flexibility of its nucleotides 6–8 and capped RNA hybridization. This would permit incorporation of UTP in position +1 of the active site. Subsequent translocation would bring the polymerase in the same state as if a capped RNA terminating by AGU is used as a primer. The 3′ template terminus would be in position −3 to −1 of the active site, ready for incorporation of an A in position 1. Following incorporation of the subsequent UTP and GTP, realignment may occur once or several times, accordingly to in vivo mRNA composition observation. This may be facilitated by the PR loop, which, by interfering with proper hydrogen bonding between the template end and the product, may reduce the number of hydrogen bonds to be broken upon realignment. Although speculative, this is supported by the absence of interaction of the extreme 3′ A nucleotide of the template with its complementary U of the capped RNA in position -3 of the active site (Fig. 6b). As for replication, we propose that (i) the destabilization of the distal duplex, (ii) the extreme tension of nucleotides 6–8, (iii) and the PR constrains may provide enough energy to trigger template repositioning. Following realignment(s) (if realignment occurs) the PR loop would retract, thereby switching the polymerase to an elongation mode. The proposed models provide hints on how replication and transcription may occur in other families of the *Bunyavirales* order[4–6,30,31] and complements the existing knowledge on prime-and-realign mechanisms used by Influenza for the replication of vRNA from a cRNA template[19,25], and for transcription[32].

In conclusion, the combination of in vitro assays and cryo-EM structural snapshots of specific LACV-L active states unveils the precise mechanisms of initiation by prime-and-realign and elongation of both replication and transcription. These results are novel for the entire *Bunyavirales* order and shed light on processes conserved in all sNSV. They will be key to design structure-based drugs to counteract these life-threatening viruses that would target essential elements of replication and transcription activity or prevent essential conformational changes.

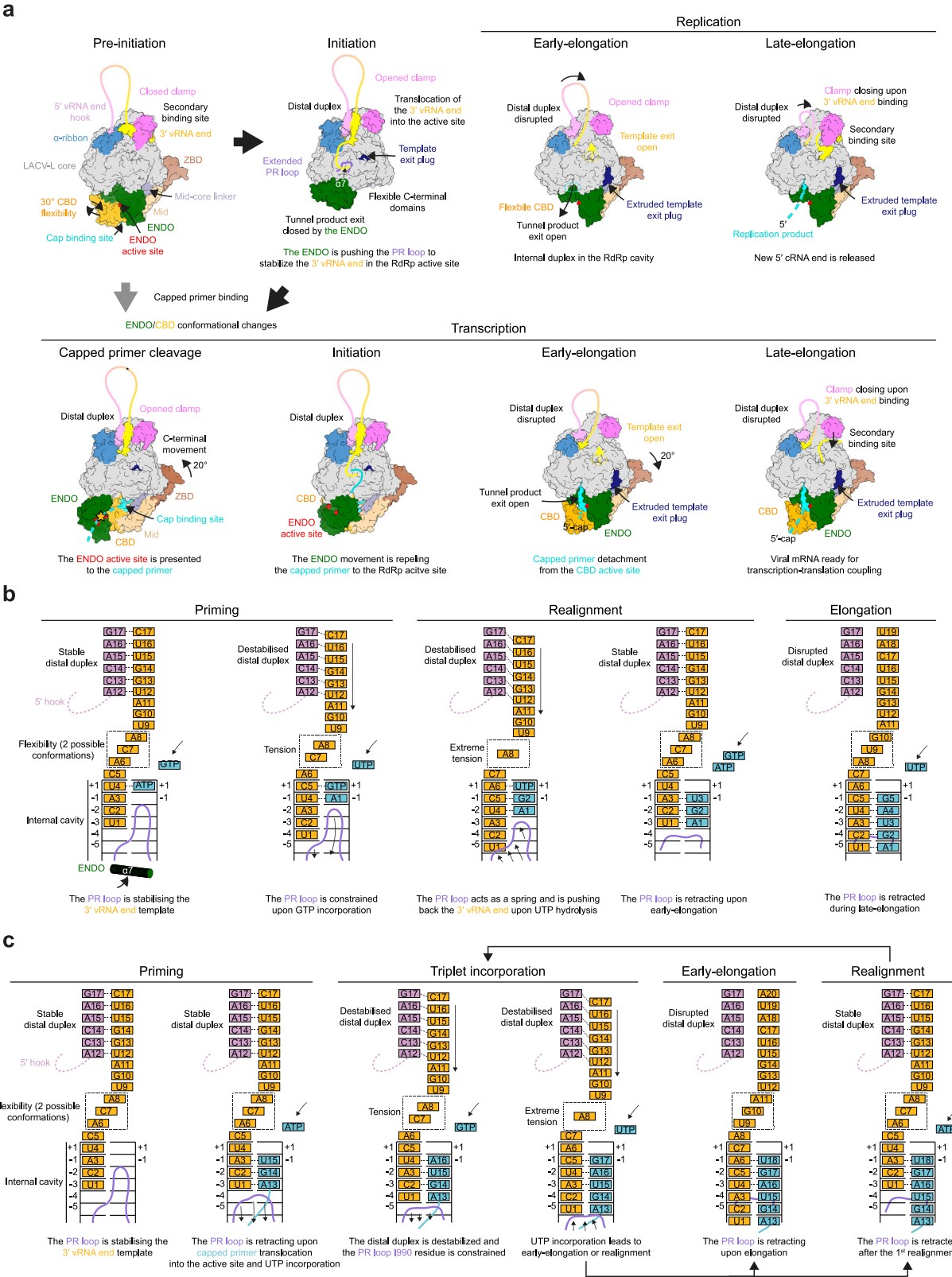

## Methods

**LACV minigenome assay**. Sub-confluent monolayers of HEK-293 cells seeded in six-well dishes were transfected with 250 ng each of plasmid pHH21-LACV-vMRen, pTM-LACV_L (or insertion mutants), pTM-LACV_N, 500 ng pCAGGS-T7, and 100 ng pTM1-FF-Luc[33,34] using Nanofectin transfection reagent in 200 μl serum-free medium (OptiMEM, Gibco-BRL). In addition, 250 ng of pI.18-HA-

PKR was added to the plasmid mix[33]. At 72 h post-transfection, cells were lysed in 200 μl Dual Luciferase Passive Lysis Buffer (Kit Promega) and both Luciferase (FF-Luc) and Renilla luciferase (REN-Luc) activities were measured as described by the manufacturer (Promega).

As pTM-LACV_L, we used N-terminal (Nter) His-tag LACV-L full-length construct (strain LACV/mosquito/1978, GenBank: EF485038.1, UniProt: A5HC98)

**Fig. 7 LACV-L replication and transcription cycle models. a** Structure-based model of the mechanisms underlying LACV-L replication and transcription. All structures are displayed as surfaces with specific domains and features colored as in Figs. 1 and 2. The vRNA is shown as a line with the 5′/3′ ends colored in pink/yellow. At initiation, the 3′-vRNA terminus is released from the secondary binding site and brought into the RdRp active site where it is stabilized by the PR loop, that extends due to the endonuclease (ENDO) movement. If the flexible cap-binding domain (CBD) snatches a 5′-cap, important conformational changes of the CBD, the ENDO, the mid and the zinc-binding domain (ZBD) occur, triggering capped primer translocation into the ENDO active site (in red). Once cleaved (yellow star), an ENDO movement triggers the capped primer repositioning towards the RdRp active site. Early elongation induces extrusion of the template exit plug resulting in the template exit opening (yellow dotted circle). In late-elongation, the 3′-vRNA template end goes back to the secondary binding site. **b, c** Structure-based model of initiation by prime-and-realign for replication (**b**) and transcription (**c**). The 5′/3′-vRNA ends are respectively colored in pink and gold. The 5′-hook structure is represented as a dotted line. Incorporated nucleotides are colored in blue. The prime-and-realign loop (PR loop) is shown in pink. The binding pocket of nucleotides A6, C7, and A8 at priming stage is surrounded by a dotted square. The proposed successive steps of the models are presented from left to right. The main role of each step is indicated below each schematic.

previously cloned into a pFastBac1 vector between NdeI and NotI restriction sites (Arragain et al, 2020), C-terminal (Cter) His-tag LACV-L full-length construct. An insertion mutant of LACV-L full-length construct (insertions of 2 internal Strep-Tag II "WSHPQFEK" flanked by a SG/GSG linkers) was as well generated between residues G1034-Y1035. The primers used to generate these constructs are in Supplementary Table 3. All reagents used are listed in Supplementary Table 4.

**Cloning, expression and purification**. The LACV-L full-length construct with an addition of one internal Strep-Tag II "WSHPQFEK" flanked by a SG/GSG linker between residues G1034 and Y1035 (G1034-SG-StrepTag II-GSG-Y1035) was chosen and recloned (Supplementary Table 3). An additional H34K mutation was inserted leading to the LACV-L$_{CItag\_H34K}$ construct used for cryo-EM analysis and activity assays. Mutations M989A/I990A/S991A (LACV-L$_{CItag\_H34K\_M989A}$/LACV-L$_{CItag\_H34K\_I990A}$/LACV-L$_{CItag\_H34K\_S991A}$) were inserted on LACV-L$_{CItag\_H34K}$ construct and used for activity assays testing the role of the PR loop (Supplementary Table 3). All LACV-L clones were made using combination of Polymerase Chain Reactions (PCR), agarose gel purification, DNA extraction using PCR clean-up kit (Machery-Nagel), Gibson assembly (NEB) and finally sequenced (Genewiz) before proceeding to further experiments.

For all LACV-L$_{CItag\_H34K}$ constructs, expressing baculoviruses were made using the Bac-to-Bac method (Invitrogen)[35]. For protein expression, High 5 (Hi5) cells at $0.5 \times 10^6$ cells/mL concentration were infected by adding 0.1% of virus and collected 72 h to 96 h after the day of proliferation arrest.

Hi5 cells were resuspended in lysis buffer (50 mM Tris–HCl pH 8, 500 mM NaCl, 2 mM β-mercaptoethanol (BME), 5% glycerol) with cOmplete EDTA-free protease inhibitor cocktail (Roche) and disrupted by sonication for 3 min (10 s ON, 20 s OFF, 50% amplitude) on ice. Lysate was clarified by centrifugation at $48,000 \times g$ during 45 min at 4 °C and filtered. Soluble fraction was loaded on pre-equilibrated StrepTrap HP column (Sigma-Aldrich) and eluted using initial lysis buffer supplemented by 2 mM d-Desthiobiotin (Sigma-Aldrich). LACV-L$_{CItag\_H34K}$ fractions were subsequently pooled, dialyzed 1 h at 4 °C in heparin buffer (50 mM Tris–HCl pH 8, 250 mM NaCl, 2 mM BME, 5% glycerol) and loaded on HiTrap Heparin HP column (Sigma-Aldrich). Elution was performed using 50 mM Tris–HCl pH 8, 1 M NaCl, 5 mM BME, 5% glycerol. LACV-L$_{CItag\_H34K}$ protein was finally loaded on a pre-equilibrated Superdex 200 Increase 10/300 GL (Sigma-Aldrich) size exclusion chromatography column in 50 mM Tris–HCl pH 8, 400 mM NaCl, 5 mM BME. Best fractions were pulled, concentrated using Amicon Ultra 10 kDa (Merck Millipore), flash frozen in liquid nitrogen and conserved at −80 °C for future experiments.

**In vitro transcription, capping, and RNAse T1 cleavage**. In vitro transcription was realized using a DNA oligo enclosing the T7 promoter sequence with an additional G at the 3′ end (5′-TAA TAC GAC TCA CTA TAG G-3′) and a template DNA oligo enclosing both complementary (i) T7 promoter sequence and (ii) 3′-5′ DNA sequence of the desired 5′-3′ RNA (Supplementary Table 3). The two nucleotides in 5′ of the primers were 2′-O methylated in order to reduce nontemplated nucleotide addition at the 3′terminus of RNAs transcribed with T7 RNA polymerase. Different templates were used to produce a 37-mer RNA (5′-GGA UGC UAU AAU AGU AGU GUA CUA CCA AGU AUA GAU A-3′), a 21-mer RNA (5′-GGA UGC UAU AAU AGU AGU GUA-3′), a 14-mer RNA (5′-GGA UGC UAU AUA GU-3′).

T7 promoter (15 µM) and template DNA (10 µM) oligos were mixed in transcription buffer (30 mM TRIS-HCl pH 8, 20 mM MgCl₂, 0.01% Triton X-100, 2 mM Spermidine), annealed at 80 °C for 2 min and cooled down to room temperature (RT). Hybridized DNA oligos (1 µM) were then added to transcription buffer supplemented by 5 mM ATP, UTP, GTP and CTP (Sigma-Aldrich), 10 mM dithiothreitol (DTT), 1% PEG8000, 50 µg/mL T7 RNA polymerase and incubated overnight at 37 °C. In vitro transcription reactions were stopped by adding 2X RNA loading dye (95% formamide, 1 mM EDTA, 0.025% SDS, 0.025% bromophenol blue, 0.01% xylene cyanol), heated 5 min at 95 °C and loaded on a 20% Tris-Borate-EDTA-7M urea-polyacrylamide gel. The corresponding products bands were cut, soaked in 0.3 M NaOAc pH 5.0–5.2 and put at rocking overnight at 4 °C. After

filtration, RNAs solutions were supplemented with 3 V of 100% ethanol, vortexed and put 2 h at −80 °C. Precipitated RNAs were pelleted by centrifugation at $48,000 \times g$ during 20 min at 4 °C. The supernatant was discarded and RNAs pellets were carefully washed with 70% ethanol before another centrifugation at $48,000 \times g$ during 20 min at 4 °C. After two consecutive washes, 70% ethanol was removed, RNAs pellets were dried 10 to 15 min before being resuspended in 50 mM TRIS-HCl pH 8 and stored at −20 °C.

In vitro RNA capping was achieved using RNAs produced by in vitro transcription as substrate and the Vaccinia capping system (NEB). The 37-mer (50 µM), the 21-mer (50 µM) and the 14-mer (20 µM) RNAs were mixed with Vaccinia Capping Enzyme buffer (NEB), 2 mM SAM, 3 pM GTP α-³²P (PerkinElmer), 10 Units of VCE and incubated 30 min at 37 °C.

To produce capped molecular markers, 2X RNA loading dye was added to capped RNA and solution was heated 5 min at 95 °C before loading on a 20% TBE-Urea-Polyacrylamide gel. Visible band corresponds to capped 37-mer (5′-m7Gppp GGA UGC UAU AAU AGU AGU GUA CUA CCA AGU AUA GAU A-3′), a capped 21-mer (5′-m7Gppp GGA UGC UAU AAU AGU AGU GUA-3′) and a capped 14-mer (5′-m7Gppp GGA UGC UAU AUA GU-3′). Additionally, capped 37-mer was digested using RNase T1 cleavage kit (Ambion). Cleavage reaction was performed for 15 min at 37 °C, stopped by adding 2X RNA loading dye and heating 5 min at 95 °C. Observed bands corresponds to a 34-mer, a 29-mer, a 19-mer, a 17-mer, and a 14-mer RNA (5′-m7Gppp GGA UGC UAU AAU AG/U AG/U G/UA CUA CCA AG/U AUA G/AU A-3′).

**Replication and transcription activity assay**. Synthetic RNAs (Microsynth) 5′-vRNA1-17 (5′-AGU AGU GUG CUA CCA AG-3′), 5′-1-17BPm (5′-ACG AGU GUC GUA CCA AG-3′), 3′-vRNA1-25 (5′-UAU CUA UAC UUG GUA GUA CAC UAC U-3′), 3′-vRNA1-30 (5′- AAC GUU AUC UAU ACU UGG UAG UAC ACU AC U-3′) were used as viral RNAs (vRNA). Capped 14-mer ending by "AG" (5′-m7Gppp GAA UGC UAU AAU AG-3′) (TriLink Biotechnologies) and in vitro produced capped 14-mer ending by "AGU" (5′-m7Gppp GGA UGC UAU AUA GU-3′) were used as capped primers.

For replication activity assays, 0.6 µM LACV-L$_{CItag\_H34K}$ were mixed with (i) 0.9 µM 5′-vRNA for 1 h at 4 °C, then 0.9 µM 3′-vRNA for overnight incubation at 4 °C.

Reactions were launched at RT, 27 °C, 30 °C, 37 °C for 0.5, 1, 2, 3, and 4 h by adding all NTPs or only ATP/GTP/UTP at 100 µM/NTP, 0.75 µCi/ml α-³²P GTP and MgCl₂/MnCl₂ in a final assay buffer containing 50 mM TRIS-HCl pH 8, 150 mM NaCl, 5 mM BME, 100 µM/NTP, 0-1-2-3-4-5 mM MgCl₂/MnCl₂. Decade markers system (Ambion) was used as molecular weight ladder.

For transcription activity assays, same conditions were used as in replication activity assay except that 0.9 µM capped RNA primer was added at the same time as the 5′-1-17BPm.

The 37-mer capped primer, before and after RNAse T1 cleavage, was used as molecular weight ladder.

Reactions were stopped by adding 2X RNA loading dye, heating 5 min at 95 °C and immediately loaded on a 20% TBE-7M urea-polyacrylamide gel that was run 2 h at 20 W. The gel was exposed on a storage phosphor screen and read with an Amersham Typhoon scanner.

In order to see if LACV-L$_{CItag\_H34K}$ replication products are 5′ triphosphate or 5′ monophosphate, a replication reaction performed as described above at 30 °C for 4 h was subjected to an incubation with 1U of Terminator Exonuclease (Lucigen) during 1 h at 30 °C in the Terminator Exonuclease buffer. The resulting product was run next to the equivalent reaction untreated with Terminator Exonuclease on a 20% Urea-PAGE gel (Supplementary Fig. 1e).

Quantification of Fig. 2e was performed based on three gels done with distinct samples. Measurements were done using ImageJ[36] and statistical analysis was done using R[37]. A two-tailed unpaired t-test was performed. P-value under 0.05 are considered significant.

All uncropped gels are presented in a source data file.

**RNA-sequencing**. To confirm the identity of the products, RNA sequencing was performed. First, in vitro replication and transcription assays were performed by

incubating at 30 °C 0.6 μM of LACV-L_CItag_H34K, 0.9 μM of capped 14-mer "AG" (for transcription only), 0.9 μM 5′-1-17BPm, 0.9 μM 3′-vRNA1-25, 100 μM of ATP/GTP/UTP/CTP in a buffer containing 50 mM TRIS-HCl pH 8, 150 mM NaCl, 5 mM BME. Reaction time was 30 min for transcription and 4 h for replication. After completion of the reaction, proteins were removed by Monarch RNA clean-up kit (NEB). Replication reactions were subjected to RNA 5′ pyrophosphohydrolase (RppH, NEB) and to a second purification with the Monarch RNA clean-up kit (NEB) to obtain purified 5′-monophosphate replication products.

RNA integrity and concentration were checked using the RNA Pico 6000 Assay Kit of the Bioanalyzer 2100 system (Agilent Technologies, Santa Clara, CA). Small RNA-Seq libraries were prepared manually from 10 ng of RNA as input using the SMARTer® smRNA-Seq kit for Illumina® Platforms (Takara Bio). Library preparation was done following the manufacturer's instructions. The size distribution of the libraries was assessed on Bioanalyzer with a DNA High Sensitivity kit (Agilent Technologies), and concentration was measured with Qubit® DNA High Sensitivity kit in Qubit® 2.0 Flurometer (Life Technologies). Libraries that passed the QC step were pooled in equimolar amounts and final pool was purified with SPRI select beads (Beckman Coulter) in a 1.3x ratio. 10 pM solution of the pool of libraries was loaded on the Illumina sequencer MiSeq and sequenced uni-directionally, generating ~17 million reads, each 60 bases long.

Raw sequencing data was processed by BCL-Convert (version 3.7.5, Illumina, San Diego, USA). After basecalling the reads were trimmed using Cutadapt (version 3.4[38]) using parameters:'-u 3 -a "AAAAAAAAAA" -e 0.3 -n 3′. Trimmed reads were then subjected to series of pattern-based searches using NR-grep (version 1.1.2[39]), GNU grep (version 2.20), and GNU AWK (version 4.0.2). Reads were grouped into categories based on the pattern found in the sequence and counted. Detailed description of the patterns can be found in Supplementary Table 5.

**Electron microscopy**. To catch structural snapshots of LACV-L_CItag_H34K in replication early-elongation state, 1.3 μM LACV-L_CItag_H34K were sequentially incubated for 1 h at 4 °C with (i) 2.6 μM 5′-1-17BPm, (ii) 2.6 μM 3′-vRNA1-25. LACV-L_CItag_H34K bound to vRNAs was subsequently incubated for 4 h at 30 °C in a final buffer containing 50 mM TRIS-HCl pH 8, 150 mM NaCl, 5 mM MgCl_2, 5 mM BME, and 100 μM of each NTP (ATP/GTP/UTP).

To catch structural snapshots of LACV-L_CItag_H34K in replication late-elongation, 1.7 μM LACV-L_CItag_H34K were sequentially incubated for 1 h at 4 °C with (i) 1.9 μM 5′-1-17BPm, (ii) 1.9 μM 3′-vRNA1-30. LACV-L_CItag_H34K bound to vRNAs was subsequently incubated for 4 h at 30 °C in a final buffer containing 50 mM TRIS-HCl pH 8, 150 mM NaCl, 5 mM MgCl_2, 5 mM BME, and 100 μM of each NTP (ATP/GTP/UTP/CTP).

To catch structural snapshots of LACV-L_CItag_H34K in transcription, 1.3 μM LACV-L_CItag_H34K were sequentially incubated for 1 h at 4 °C with (i) 1.9 μM 5′-1-17BPm and 3.9 μM commercial 14-mer capped primer finishing by "AG", (ii) 1.9 μM 3′-vRNA1-25. LACV-L_CItag_H34K bound to vRNAs and capped primer was subsequently incubated for 1 h at 30 °C in a final buffer containing 50 mM TRIS-HCl pH 8, 150 mM NaCl, 2 mM MgCl_2, 5 mM BME, and 100 μM of NTPs (ATP/UTP) for transcription cleavage/capped primer entry/ initiation states or 100 μM of each NTP (ATP/GTP/UTP) for transcription early-elongation state.

Both replication and transcription reactions were put at 4 °C before grids freezing. For each reaction, 3.5 μl of the sample were deposited on previously glow-discharged (25 mA, 45 s) UltraAuFoil gold grids 300 mesh, R 1.2/1.3. Excess solution was blotted for 2 s, blot force 1, 100% humidity at 20 °C with a Vitrobot Mark IV (Thermo Fischer Scientific) before plunge-freezing in liquid ethane.

Automated data collections of (i) the replication early-elongation state, (ii) the replication late-elongation state, (iii) the transcription initiation state, and (iv) the transcription early-elongation state were performed on a 200 kV Glacios cryo-TEM microscope (Thermo Fischer Scientific) equipped with a K2 direct electron detector (Gatan) using SerialEM[40]. Coma and astigmatism correction were performed on a carbon quantifoil grid. Movies of 60 frames were recorded in counting mode at a 36,000× magnification giving a pixel size of 1.145 Å with defocus ranging from −0.8 to −2.0 μm. Total exposure dose was 60 e−/Å². The number of movies per experiment are indicated in Supplementary Figs. 3, 4, and 8 and in Supplementary Tables 1 and 2.

To improve the resolution of the (i) replication initiation state, the (ii) transcription capped primer entry state and the (iii) transcription initiation state, an automated data collection was performed on a Titan Krios G3 (Thermo Fischer Scientific) operated at 300 kV equipped with a K3 (Gatan) direct electron detector camera and a BioQuantum energy filter using SerialEM. Micrographs were recorded in counting mode at a ×130,000 magnification giving a pixel size of 0.645 Å with defocus ranging from −0.8 to −1.8 μm. Movies of 40 frames were collected with a total exposure of 50 e−/Å². In all, 15573 movies were collected.

**Image processing**. For each collected dataset, movie drift correction was performed in Motioncor2[41]. For images collected on the Thermo Fischer Scientific Glacios TEM, the two first frames and the last ten were removed. For images collected on the Thermo Fischer Scientific Titan Krios TEM, only the first two first frames were removed. Both gain reference and camera defect corrections were applied. Further initial image processing steps were performed in cryoSPARC v3.2.0[42]. CTF parameters were determined using "Patch CTF estimation" on non-dose weighted micrographs. Realigned micrographs were then inspected and low-

quality micrographs displaying crystalline ice, ice contamination or aggregates for example were manually discarded for further image processing. LACV-L_CItag_H34K particles were picked using a circular blob with a diameter ranging from 90 to 170 Å. The number of particles picked is indicated in Supplementary Figs. 3, 4, and 8 and in Supplementary Tables 1 and 2. For Thermo Fischer Scientific Glacios datasets, particles were extracted from dose-weighted micrographs using a box size of 260 × 260 pixels². For the Thermo Fischer Scientific Titan Krios dataset, the box size was 440 × 440 pixels². For each dataset, the same image processing approach was used to separate the different LACV-L conformation and get the best map quality. First, successive 2D classifications were used to eliminate bad quality particles displaying poor structural features. LACV-L_CItag_H34K initial 3D reconstruction was generated with "Ab-initio reconstruction" in cryoSPARC using a small subset of particles. All selected particles were subjected to 3D refinement with per-particle CTF estimation. The rest of the image processing was done in RELION 3.1[43,44]. For each dataset, particles were divided in equal subset of 250k particles and subjected to multiple 3D classification (10 classes for each subset) with coarse image-alignment sampling using a circular mask of 170 Å. For each identical LACV-L conformation, particles were grouped and subjected to a 3D refinement with a circular mask of 170 Å followed by 3D classification using local angular searches based on previously determined orientations. High-resolution classes and particles belonging to the same conformation were finally grouped and subjected to 3D masked refinement with local angular searches to obtain high-resolution structure. Masks to perform sharpening were generated using last 3D refined maps which has been low-pass filtered at 10 Å, extended by 4 pixels with 8 pixels of soft-edge. Post-processing was done using a manually determined B-factor. For each final map, reported global resolution is based on the FSC 0.143 cut-off criteria and local resolution variations were also estimated in RELION 3.1 (Supplementary Figs. 3, 4, and 8).

**Model building in the cryo-EM maps**. Both LACV-L in pre-initiation state (PDB: 6Z6G) and LACV-L in elongation mimicking state (PDB: 6Z8K) models[12] were used as a starting point to manually build into the different cryo-EM maps in either replication/transcription initiation/elongation states using COOT[45]. All the RNAs (5′-1-17BPm, 3′-vRNA1-25, 3′-vRNA1-30, the capped primer and both replication and transcription products) were manually built using COOT. Each model was refined using Phenix-real space refinement[46]. Atomic model validation was performed using the Phenix validation tool and the PDB validation server. Model resolution according to cryo-EM map was estimated at the 0.5 FSC cutoff. Figures were generated using ChimeraX[47]. Electrostatic potential was calculated using PDB2PQR[48] and APBS[49].

**Multiple alignment**. Multiple alignment was performed using Muscle[50] and is displayed using ESPript[51].

**Reporting summary**. Further information on research design is available in the Nature Research Reporting Summary linked to this article.

## Data availability

The data that support this study are available from the corresponding authors upon reasonable request. The coordinates and EM maps generated in this study have been deposited in the Protein Data Bank and the Electron Microscopy Data Bank. LACV-L replication initiation state 7ORN and EMD-13043; LACV-L replication early-elongation state: 7ORO and EMD-13044; LACV-L replication late-elongation state: 7ORI and EMD-13038; LACV-L transcription capped primer cleavage state: 7ORJ and EMD-13039; LACV-L transcription capped primer active site entry state: 7ORK and EMD-13040; LACV-L transcription initiation state: 7ORL and EMD-13041; LACV-L transcription early-elongation state PDB 7ORM and EMD-13042. Previously released coordinates and EM maps used in this study: LACV-L pre-initiation state (6Z6G and EMD-11093); LACV-L elongation-mimicking state (6Z8K and EMD-11118). The NGS raw reads generated in this study have been deposited in the European nucleotide archive under accession code ERP132950. Source data are provided with this paper.

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

## Acknowledgements

We thank Karine Huard, Angélique Fraudeau, Petra Drncová, Alice Aubert and Martin Pelosse for technical advices on expression and purification; Friedemann Weber for providing the LACV mini-replicon to S.C.; Sissy Kalayil, Tomas Kouba, Anna Duban-kova, and Joanna Wandzik for technical advices on radioactivity experiments; Vladimir Benes from EMBL Heidelberg GeneCore for NGS; Lionel Imbert and Alice Stelfox for advices on in vitro transcription; Simon Fromm for data collection on the Heidelberg Titan Krios; Aymeric Peuch for setting up and maintaining the EM computing cluster; Piotr Gerlach and Juan Reguera for useful discussions and Daphna Fenel for technical support. This work used the platforms of the Grenoble Instruct-ERIC center (ISBG; UAR 3518 CNRS-CEA-UGA-EMBL) within the Grenoble Partnership for Structural Biology (PSB), supported by FRISBI (ANR-10-INSB-05-02) and GRAL, financed within the University Grenoble Alpes graduate school (Ecoles Universitaires de Recherche) CBH-EUR-GS (ANR-17-EURE-0003). The electron microscope facility is supported by the Auvergne-Rhône-Alpes Region, the Fondation pour la Recherche Médicale (FRM), the fonds FEDER and the GIS-Infrastructures en Biologie Santé et Agronomie (IBiSA). This work benefited from access to the cryo-electron microscopy platform of the European Molecular Biology Laboratory (EMBL) in Heidelberg and has been supported by iNEXT-Discovery, project number 871037, funded by the Horizon 2020 program of the European Commission. We thank all platform staff that enabled us to perform these analyses. IBS acknowledges integration into the Interdisciplinary Research Institute of Grenoble (IRIG, CEA). This work was supported by the ANR-19-CE11-0024-02 and the Institut Universitaire de France endowment to H.M., the FRM grant number FDT202012010396 to B.A.

## Author contributions

F.B. and B.A. cloned LACV-L FL with internal tags. F.B. performed mini-replicon assays. B.A. and Q.D.T. cloned mutant constructs. B.A. and Q.D.T. expressed and purified LACV-L. B.A. performed in vitro activity assays. B.A. and H.M. prepared cryo-EM grids. H.M. and B.A. collected cryo-EM data on a Thermo Fischer Scientific Glacios EM thanks to advices and training from G.S. who set up and maintains the IBS-ISBG EM platform.

B.A. performed cryo-EM image processing. B.A. and H.M. built the models based on the cryo-EM maps with inputs from S.C.; B.A. and H.M. performed structural analysis. N.A. performed RNA library preparation and NGS. J.P. performed bioinformatic analysis of NGS. H.M. and G.S. co-supervise B.A. and Q.D.T. The project was conceived by H.M. with inputs from S.C. and G.S. This project used funding obtained by H.M. and G.S. The manuscript was written by H.M. and B.A. with input from all authors.

## Competing interests

The authors declare no competing interests.
