## [Peer Review File · Nature Communications]

Structural snapshots of La Crosse virus polymerase reveal the mechanisms underlying Peribunyaviridae replication and transcriptionReviewers' Comments:

Reviewer #1:

Remarks to the Author:

The major goals of this paper were to solve structures of transcription and replication intermediates of the La Crosse virus polymerase L. By comparing these states to an established in vitro transcription and replication assay, the authors sought to provide new mechanistic insights into the function of the polymerase. The major success of this paper is the capture of these intermediate states and the attempted categorization of these states into the replication and transcriptional cycles of this enzyme. The major weaknesses of this paper are the lack of figures showing the actual electron density of these different complexes which would allow better evaluation of the states as described and the interpretability of the in vitro assays. This paper builds upon previous structures of the La Crosse virus polymerase and other related viruses polymerases and provides the first snapshots of intermediates within the transcription and replication processes.

Major points:

1. I have a hard time interpreting the in vitro transcription and replication assays because the molecular weight marker indicated on the gel does not match the reported sizes of the product bands on the gel. If the authors are able to validate that the products are what they say they are that would go a long way towards validating the conclusions drawn in the paper.
2. No where in the paper do the authors show the quality of the electron density maps themselves to validate primarily the sequences and register of the RNA strands in the structures. This would go a long way to validate that the states that are described in the paper accurately reflect the structures that were solved.
3. I found it difficult to follow the naming of the PR loop when the authors refer to a "priming loop" (line 187). Is there a conserved "priming loop" that is not serving that role in La Crosse virus and is replaced by the PR loop?
4. There is a lot of redundancy of text between the discussion and the results section. I would recommend reducing the discussion significantly.

Minor points:

1. Line 75 "to a construct without a tag"
2. Line 76-77 "to the LACV-L without a tag,"
3. Line 78 "To abolish unspecific RNA degradation in vitro,"
4. Line 79 "Despite the optimized construct"
5. Line 83 "as single-stranded RNA at their"
6. Line 86 "we analyzed"
7. Line 88 — introduce 5'-1-17Bpm here
8. Line 116 "elongation at 2.8, 2.9, and 3.9 Å resolution, respectively."
9. Line 121 — Is UTP required in this instance? Why was it included if the complex is stalling with an ATP present.
10. Line 127 and elsewhere — extremity is not the correct word choice here. Try "end" or "terminus" instead.
11. Line 129 "changes of LACV-L during initiation compared to the stable conformation found in the pre-initiation"
12. Line 132 "entire CTER, including the CBD,"
13. Line 144 — it's unclear how the data supports the conclusion that A6, C7, and A8 are less coordinated and adopt different conformations.
14. Line 147 — please state which canonical polymerase motifs since it is unclear what is being referred to.
15. Line 153 "proximal to the active site, triggering the PR loop to extend." — What role does the ENDO play in this? It's unclear in Fig 2C that there are stabilizing interactions that would implicate the ENDO driving this rearrangement.
16. Line 157 — Which depicted map are the authors referring to here? Showing the density for this

ATP would be beneficial.

17. Figures 2D and 2E should be switched to reflect the order they are referenced in the text.
18. Line 164 — What structural role is M989 serving that results in the abolition of replication activity when mutated?
19. Line 167-168 “We speculate... the product, facilitates elongation.”
20. Line 168-170 — this sentence needs rewording
21. Line 171 “3'-vRNA, this residue appears to be important for proper”
22. Figure 3a shows many structures but only one is mentioned in the text.
23. Line 176 “repositioning, provides space for the initial”
24. Line 178 “assay results in LACV-L stalling with G10 in”
25. Line 179-181 — I could not understand what this sentence is saying.
26. Line 182-184 — It's unclear that this movement is necessary to accommodate the RNA duplex and not just the result of stabilization of the helix by the template RNA.
27. Line 188 — “renamed”
28. Line 198-199 — The data provided does not clearly show that the movements that are described here result in the 3'-vRNA entering into the template entry tunnel.
29. Line 208 “chosen capped RNA has”
30. Line 210 “Time, divalent ion identity, and divalent ion concentration were... after 30 minutes in the presence”
31. Line 213 “products, a 37-/36-mer”
32. Line 215-216 “hybridization of the last 2 or 3 template nucleotides”
33. Line 221 “Usage of an NTP”
34. Line 231 “processing enabled us to”
35. Line 235 — It is hard from the figure to tell that the capped RNA is oriented towards the ENDO cleavage site, it looks to me like the other direction.
36. Line 239-240 — It would help to use consistent conformation names between the text and figures. No where is “capped primer active site entry” found in Fig 4.
37. Line 245-247 — reword this sentence, it is difficult to read as is.
38. Line 254 “ENDO movement resulting in a 160” — please also label this in fig. 4e
39. Line 256 “In addition to the ENDO movement” — label thumb in fig. 5a
40. Line 265 — It's hard to see this rotation in the figures themselves, might help to redesign the figures to highlight this movement.
41. Line 281 — what makes cap binding peculiar?
42. Line 293 “Because these residues bind to the phosphate backbone, it is likely that binding is sequence independent”
43. Line 296 “indicating tighter binding”
44. Line 300 — It is confusing what early-state initiation refers to.

Reviewer #2:

Remarks to the Author:

Arragain et al have captured various elongating LACV RNA polymerase complexes. To obtain these structures, which had proved difficult so far, the authors changed the position of the tag on the polymerase, introduced a number of clever mutations in the promoter strands, and mutated the endonuclease domain of the RNA polymerase to prevent inadvertent cleavage of products.

One of the most exciting insights from these new structures is how prime-realignment may be achieved during negative strand virus replication and transcription. Prime-realignment is a process in which the RNA polymerase of negative strand RNA viruses copies 2-3 nucleotides once (or more) before starting processive elongation. This process has been observed for a number of RNA viruses, including the influenza A virus.

Through their new structures, Arragain et al link the prime-realignment mechanism to a novel

structural element, which they call the PR loop. Mutation of this loop abrogates or increases RNA polymerase activity, underlining that the loop is important for polymerase activity. The authors also note that one of the mutations leads to the formation of products that migrate slower in denaturing PAGE, suggesting that there is a link between the function of the loop and the stimulation of prime-realignment. This part would be strengthened by sequencing data of e.g. RNA extracted from the mini-genome assays with the wt and mutant RNA polymerases.

Overall, I think that the data are presented with great clarity, that the conclusions are well-supported by the data, and that the insights from this study are important to the field. The data and insights are also in line with previous studies on the influenza A virus RNA polymerase, where mutation of a comparable loop also affected realignment. I have a couple of points that would strengthen the authors' manuscript.

1. Have the realignment products from mini-replication assays been sequenced to confirm a) that the mutants are triggering more realignment events in cell culture and b) that the slower migrating bands identified as realignment products in PAGE are truly realignment products?
2. It would be helpful if the gel data in Fig. 2e were quantified and statistics performed.

Dear Referees,

We thank you for your positive appreciations and constructive comments on our manuscript entitled “Structural snapshots of La Crosse virus polymerase reveal the mechanisms underlying *Peribunyaviridae* replication and transcription”. Please find below our point-by-point responses.

Reviewer #1 (Remarks to the Author):

The major goals of this paper were to solve structures of transcription and replication intermediates of the La Crosse virus polymerase L. By comparing these states to an established in vitro transcription and replication assay, the authors sought to provide new mechanistic insights into the function of the polymerase. The major success of this paper is the capture of these intermediate states and the attempted categorization of these states into the replication and transcriptional cycles of this enzyme. The major weaknesses of this paper are the lack of figures showing the actual electron density of these different complexes which would allow better evaluation of the states as described and the interpretability of the in vitro assays. This paper builds upon previous structures of the La Crosse virus polymerase and other related viruses polymerases and provides the first snapshots of intermediates within the transcription and replication processes.

Major points:

1. *I have a hard time interpreting the in vitro transcription and replication assays because the molecular weight marker indicated on the gel does not match the reported sizes of the product bands on the gel. If the authors are able to validate that the products are what they say they are that would go a long way towards validating the conclusions drawn in the paper.*

Concerning replication assays, we use the decade marker, which is a standard practice as shown in recent articles focusing on viral polymerase replication activity. As examples, we can cite (i) Vogel *et al.*, JBC, 2019, that shows Lassa-L replication assays, (ii) Vogel *et al.*, NAR, 2020, that focuses on SFTSV-L replication assays and (iii) te Velhuis *et al.*, J Virol, 2018, that gives insights into influenza polymerase replication. To have a more precise ladder and validate the position labelled on the main gels, we have now included a complementary experiment shown in the newly incorporated **Supplementary Fig. 1b**. It shows that the 25-mer template (3'vRNA1-25) migrates as the product that we initially labelled as being a 25-mer product, thereby validating our initial labelling.

In addition to clarifying the position on the gels, we have performed RNA Next Generation Sequencing (NGS) to validate the nature of the products. NGS was performed on a reaction done in presence of LACV-L_{Citag_H34K}, 5'-1-17Bpm, 3'vRNA1-25 and 4 nucleotides. This approach confirms the presence of the expected 25-mer product. It also identifies a misincorporation in the terminal 5' nucleotide for some 25-mer products, that would migrate as the expected 25-mer product on gels. In addition, NGS identifies the presence of

“realigned products”, as proposed in the initial version of the manuscript. These products that are extended in 5’ with GU and GGU sequences.

These new results are now mentioned in the main text:

“Next Generation RNA sequencing (NGS) confirms the presence of expected 25-mer replication products (**Supplementary Fig. 2a**). It also identifies 25-mer products that contain a misincorporated terminal 5’ nucleotide and therefore start with GGU instead of AGU. Slower-migrating products are also visible on gel and NGS identifies them as being products extended in 5’ with GU and GGU sequences.”

Concerning transcription assays, we showed in the initial version of the manuscript a 37-mer molecular weight marker that had the same sequence as the transcription product, already clearly showing in **Fig. 4 a and b** that the expected transcription product had the correct size. We have pursued this strategy and have obtained a molecular weight marker with a correct size/composition for another product: the reaction with LACV-L_{Citag_H34K}, 5’-1-17BPm, 3’vRNA1-25 and 3 nucleotides. In this case, the expected product is a 21-mer. The added molecular weight marker clearly confirms that the product observed has the expected size. This is now shown in **Supplementary Fig 7c**.

In addition, we also performed NGS of the reaction with LACV-L_{Citag_H34K}, cap14AG, 3’-vRNA1-25, 5’-1-17BPm and 4 nucleotides. This confirms that the 37-mer band is indeed what we were reporting in the article. It also reveals the presence of primed-and-realigned transcripts that result in the incorporation of AGU nucleotide triplet(s) (once, twice or three times) after the primer before proceeding to elongation of the product. This is now indicated in the main text:

“The minority 37-/36-mer products correspond to the capped RNA size elongated by 23/22 nucleotides, considering the hybridization of the last 2 or 3 template nucleotides with the cap14AG/capAGU. Their migration is consistent with a capped molecular weight marker of the same size and composition, and they are identified in NGS (**Fig. 4a, b**). The majority 40-/39-mer products correspond to primed and subsequently realigned transcripts, that result in the addition of an AGU nucleotide triplet at the end of the capped primer before elongation, as detected by NGS analysis (**Supplementary Fig. 2b**). NGS also reveals the presence of products that have been realigned up to three times.”

Altogether, the molecular weight markers and the RNA-sequencing explain the length and the nature of the products. They confirm what was proposed in the initial manuscript and complement it, clearly adding value to the results. We would therefore like to thank Reviewer 1 for this constructive comment.

2. No where in the paper do the authors show the quality of the electron density maps themselves to validate primarily the sequences and register of the RNA strands in the structures. This would go a long way to validate that the states that are described in the paper accurately reflect the structures that were solved.

We have added two Supplementary Figures that clearly show the quality of the electron density: **Supplementary Figure 5** for the replication maps and **Supplementary Figure 9** for the transcription maps. For clarity, only the RNA model and the corresponding density are shown in these Supplementary Figures. Zooms are made on specific parts that unambiguously show the register of RNA strands, clearly confirming that the states described in the paper accurately reflect the structures that were solved.

In addition, as we agree it is crucial for Reviewers to clearly visualize that the models correspond to the density maps, we have given access them to all the maps and models (this was done at the initial article submission). These maps and models are also deposited in the

PDB and EMDB and will therefore be available to all the scientific community as soon as the paper is published. The reviewers also have access to PDB and map validation documents.

3. I found it difficult to follow the naming of the PR loop when the authors refer to a “priming loop” (line 187). Is there a conserved “priming loop” that is not serving that role in La Crosse virus and is replaced by the PR loop?

Yes, Reviewer 1 exactly points what we see and what is described in the text.

Several viral polymerases that perform *de novo* initiation have a loop called the “priming loop” that plays a major role in *de novo* replication initiation by stabilizing the first nucleotides to be incorporated in the product. For example, in Dengue, Hepatitis C or Influenza viruses, this loop protrudes from the thumb domain of the polymerase. LACV-L presents such a loop in the expected position of the thumb domain and performs *de novo* replication. In the original article on LACV-L structure (Gerlach et al, Cell, 2015), this loop was therefore called “priming loop”, even though it was disordered in the described pre-initiation state. In the replication initiation structure of the current article, we see that the loop appears too far from the active site to be directly implicated in replication initiation. We therefore renamed this loop “template exit plug” as it is changing its position in elongation to enable the template to exit the active site cavity by the template exit channel.

In the present manuscript, we discover another loop that we call the “prime-and-realign loop”, PR loop. The importance of this loop in prime and realign is assessed by (i) its conformational change between replication initiation and elongation, (ii) the impact of mutation of this loop in replication initiation/realignment and in transcription realignment. As mentioned by Reviewer 2, the discovery of the PR loop is one central element of the current manuscript.

4. There is a lot of redundancy of text between the discussion and the results section. I would recommend reducing the discussion significantly.

We have now reduced the discussion. In particular, we don't discuss the importance of the distal duplex anymore, we just refer to articles related to this specific point. We need here to express the difficulty to make the discussion shorter due to the fact Reviewer 1 brings several points that necessitate adding information (notably minor point 9 and minor point 41). To fulfill Reviewer 1 request at best, we: (i) summarized the initially submitted discussion, (ii) added the necessary information to answer minor point 9, (iii) answered minor point 41 only in the response to the Reviewers to remain concise.

As Reviewer 1 states that the discussion and the results are redundant, we clarify here why they are different. Discussion consists of 3 parts:

In the first and second parts we compare LACV-L with other sNSV polymerases. We discuss the presence, the composition, the positioning of the prime-and-realign loop in different viral polymerases. This is essential, as pointed by Reviewer 2: *“One of the most exciting insights from these new structures is how prime-realignment may be achieved during negative strand virus replication and transcription. This process has been observed for a number of RNA viruses, including the influenza A virus.”* We also discuss the difference in positioning and conformational changes of the CBD and the ENDO during transcription

between Influenza polymerase and LACV-L. Such comparisons put the results on LACV-L in the context of all the sNSV and therefore significantly increases the impact of the results. In the third part we propose complete models of replication and transcription. These results are a major advance in the field as they depict not only the entire cycle of active transcription but are the first to do the same for the distinct process of replication. This section brings together elements from the 7 different structures and from activity assays to propose a detailed mechanistic model summarized in Figure 7. Without it, the structures and the activity assays would not be put in the context of the replication and transcription cycles and the article would have much less impact. We here again would like to cite Reviewer 2, to show that the discussion is relevant in the field: "Overall, I think that the data are presented with great clarity, that the conclusions are well-supported by the data, and that the insights from this study are important to the field."

Minor points:

1. Line 75 "to a construct without a tag"

The text has been updated.

2. Line 76-77 "to the LACV-L without a tag,"

The text has been updated.

3. Line 78 "To abolish unspecific RNA degradation in vitro,"

The text has been updated.

4. Line 79 "Despite the optimized construct"

The text has been updated.

5. Line 83 "as single-stranded RNA at their"

The text has been updated.

6. Line 86 "we analyzed"

The text has been updated.

7. Line 88 — introduce 5'-1-17BPm here

We updated the text to introduce 5'-1-17BPm: "We generated a 17-base-pair modified 5' RNA (5'-1-17BPm), by mutating the nucleotides G2, U3, A9 and C10 of the 5'-end into C2, G3, C9 and G10, thereby preserving the hook structure and its interaction with LACV-L while significantly decreasing the 5'/3'-vRNA complementarity (**Fig. 1c**, pre-initiation vs initiation)

8. Line 116 "elongation at 2.8, 2.9, and 3.9 Å resolution, respectively."

The text has been updated.

9. Line 121 — Is UTP required in this instance? Why was it included if the complex is stalling with an ATP present.

UTP was included for two reasons:

- 1) ATP is seen in position +1 of the active site

- 2) polymerization reaction occurs between nucleotides present in position -1 and position +1 of the active site.

In this situation, performing the reaction in absence of UTP would have raised the question: what happens if UTP is added? The absence of UTP in position -1 appears to indicate that it is not incorporated at replication initiation. This is consistent with the fact that the replication generates triphosphate products starting with 5'pppAGU...3' (triphosphate 5' end starting with A and not U). LACV-L (*Peribunyaviridae*) replication initiation therefore appears to differ from HTNV-L replication initiation that generates monophosphate 5' RNA (Garcin *et al.*, 1995 ; Habjan *et al.* 2008). The Garcin *et al.* article suggests that to obtain 5' monophosphate an extra nucleotide would be added in 5' (the equivalent of the UTP in LACV) and then cleaved, leaving a monophosphate.

These aspects raised by Reviewer 1 minor point 9 are added in the article:

- a) The absence of UTP in position -1 is now mentioned in the result section:

“Although UTP was added in the mix, it is not observed in position -1, which appears to indicate that it is not incorporated at initiation.”

- b) Comparison of LACV-L and HTNV-L initiation model (addition of an extra nucleotide in 5' and cleavage in monophosphate) is now added in the discussion:

“The exact mechanism of initiation may however slightly differ between LACV-L and HTNV-L. Indeed, the 5' end of HTNV RNA products have been shown to be monophosphate^{4,24} whereas we show here that LACV-L products have triphosphate ends (**Supplementary Fig. 1e**). The model proposed for HTNV-L, that would need to be validated experimentally, is that an extra nucleotide in 5' is added in the prime-and-realign process, that would then be cleaved, leaving a monophosphate at the 5' end. Hantavirus PR loop might thus play an additional role at replication initiation.”

10. Line 127 and elsewhere — extremity is not the correct word choice here. Try “end” or “terminus” instead.

This request has been taken into account. “Extremity” has been replaced by either “end” or “terminus” through all the text.

11. Line 129 “changes of LACV-L during initiation compared to the stable conformation found in the pre-initiation”

The text has been updated.

12. Line 132 “entire CTER, including the CBD,”

The text has been updated.

13. Line 144 — it's unclear how the data supports the conclusion that A6, C7, and A8 are less coordinated and adopt different conformations.

The residues that interact with the 3'-vRNA template in the replication initiation state are shown on **Fig. 2b**. This is now indicated in the figure legend. It shows that A6 and C7 are less coordinated than the other nucleotides.

The density for residues A6, C7 and A8 is now shown on the added **Supplementary Fig. 5a**. This clearly shows their multiple conformations.

14. Line 147 — please state which canonical polymerase motifs since it is unclear what is being referred to.

It is mainly the motif F and the finger domain that are implicated in 3'-vRNA terminal nucleotide stabilization. This is now clearly mentioned in the text.

“The 3' terminal nucleotides U1 to C5 display clearer density correlated with their higher degree of stabilization by the finger domain and in particular the canonical polymerase motif F (Fig. 2c, Supplementary Fig. 5a).”

15. Line 153 “proximal to the active site, triggering the PR loop to extend.” — What role does the ENDO play in this? It's unclear in Fig 2C that there are stabilizing interactions that would implicate the ENDO driving this rearrangement.

The ENDO movement is large between the pre-initiation and the initiation state. In the absence of concomitant movement of the PR loop, the ENDO residues E177 and K181 would clash with the residues 983 and 984 of the PR loop. This is now indicated in the text: “At initiation, the ENDO conformational change brings its residues 172 to 184 (α -helix 7) proximally to the PR loop (Fig. 2c). The ENDO residues E177 and K181 would clash with the PR loop resting state conformation, suggesting that the ENDO movement is linked with the PR loop extension. In this extended state, the PR loop residue M989 interacts with the 3' extreme nucleotide U1 in position -3 of the active site while the residues 1990 and S991 stabilize the nucleotide C2 in position -2 of the active site (Fig. 2c).”

16. Line 157 — Which depicted map are the authors referring to here? Showing the density for this ATP would be beneficial.

We refer to the replication initiation map as indicated in the title of the paragraph. ATP density is now shown in the extra **Supplementary fig. 5a**.

17. Figures 2D and 2E should be switched to reflect the order they are referenced in the text.

These panels are presented in this order to facilitate the comparison between replication initiation (panel c) and replication elongation (panel d) for readers. These two panels indeed show the same elements in the same orientation for the two states.

18. Line 164 — What structural role is M989 serving that results in the abolition of replication activity when mutated?

As indicated in the text, “the PR loop residue M989 interacts with the 3' terminal nucleotide U1 in position -3 of the active site”. We therefore suggest that M989A abolishes the replication due to its importance “in precise template positioning”.

19. Line 167-168 “We speculate... the product, facilitates elongation.”

The paragraph related to this point has been modified to answer properly Reviewer 2 point 2. The sentence that Reviewer 1 requested to be updated is not present anymore.

20. Line 168-170 — this sentence needs rewording.

The paragraph related to this point has been modified to answer properly Reviewer 2 point 2. The sentence that Reviewer 1 requested to be updated has therefore been reworded.

21. Line 171 “3'-vRNA, this residue appears to be important for proper”

The paragraph related to this point has been modified to answer properly Reviewer 2 point 2. The sentence that Reviewer 1 requested to be updated is not present anymore.

22. Figure 3a shows many structures but only one is mentioned in the text.

We refer to Figure 3a in the sentence: “Progression towards elongation implies important remodeling of LACV-L domains. Retraction of the PR loop, that is coupled with ENDO repositioning (Fig. 2d, 3a), leads to the initial formation of the template-product duplex in the active site cavity.”

We here refer to conformational changes between initiation (one structure) and elongation (two structures). These three structures are shown **Figure 3a**.

23. Line 176 “repositioning, provides space for the initial”
The text has been updated.

24. Line 178 “assay results in LACV-L stalling with G10 in”
The text has been updated.

25. Line 179-181 — I could not understand what this sentence is saying.

The sentence to which Reviewer 1 is referring to is the following: “Whereas the replication initiation was performed internally, realignment must have occurred as the replication elongation displays an entire product with the 5'-cRNA end corresponding to nt 1.”

One important point of the article is to describe the prime-and-realign mechanism. This is firstly described in the introduction: “LACV-L and other *Peribunyaviridae* are suspected to initiate their replication internally at position 4 of the RNA template to produce a primer that then realigns to the template end. This process, called “prime-and-realign”, is made possible by a triplet nucleotide repetition at the 3'-vRNA template end (3'-UCAUCA...-5' for LACV) and has been reported for several families in the *Bunyavirales* order, although with family-dependent specificities⁴⁻⁶.”

We describe in the article an internal initiation at nucleotide 4 in the replication initiation mimicking state.

In the absence of realignment, the cRNA would therefore not be complementary to the vRNA, the 5'-cRNA end would correspond to nucleotide 4. Here we see that the cRNA is complementary to the vRNA, starting at nucleotide 1.

We have updated the sentence, to clarify this further: “Whereas the replication initiation was performed internally, realignment must have occurred as the replication elongation displays an entire product with the 5'-cRNA end that is complementary to the 3'-vRNA nucleotide 1.”

26. Line 182-184 — It's unclear that this movement is necessary to accommodate the RNA duplex and not just the result of stabilization of the helix by the template RNA.

Reviewer 1 is referring to the opening of the lid, the thumb and the thumb-ring coupled to the extension of the bridge loop that we suggest being necessary to accommodate the RNA duplex. The superposition of the replication initiation and elongation states clearly shows that the RNA duplex would clash with the lid, the thumb and the thumb-ring in replication initiation position as shown in the figure below.

Equivalent movements have been reported for influenza virus polymerase (Kouba *et al.*, NSMB, 2019, Wandzik *et al.*, Cell, 2020) and are therefore common to several segmented negative stranded RNA viruses.

27. Line 188 — “renamed”

The text has been updated.

28. Line 198-199 — The data provided does not clearly show that the movements that are described here result in the 3'-vRNA entering into the template entry tunnel.

The data that Reviewer 1 is mentioning here consists in two maps: the pre-initiation map and the initiation map. Comparison of these two maps shows:

For LACV-L: movements of the α -ribbon, vRBL, arch and clamp (**Fig. 3c**).

For the 3'-vRNA: change of its position from the 3'-vRNA secondary binding site to the template entry tunnels.

This is clearly seen in the structures. We have updated the text to make this inambiguous point clearer:

“At pre-initiation, the α -ribbon is ordered and the vRBL is proximal to the core, thereby forming a cleft called the 3'-vRNA secondary binding site to which the 3'-vRNA end binds specifically. At initiation, the acquired α -ribbon flexibility and displacement of the vRBL away from the core disrupt the 3'-vRNA secondary binding site. The 3'-vRNA changes its position to orient itself into the template entry tunnel necessary for initiation.”

29. Line 208 “chosen capped RNA has”

The correction requested is incorrect as there are two capped RNA primers. We updated the text as such: “The chosen capped RNA primers have”

30. Line 210 “Time, divalent ion identity, and divalent ion concentration were... after 30 minutes in the presence”

The text has been updated.

31. Line 213 “products, a 37-/36-mer”

The text has been updated.

32. Line 215-216 “hybridization of the last 2 or 3 template nucleotides”

The text has been updated.

33. Line 221 “Usage of an NTP”

The text has been updated.

34. Line 231 “processing enabled us to”

The text has been updated.

35. Line 235 — It is hard from the figure to tell that the capped RNA is oriented towards the ENDO cleavage site, it looks to me like the other direction.

Only two nucleotides of the capped RNA are visible. We disagree that the capped RNA is oriented in the other direction, but we agree that it remains speculative that the conformation we see is the “endonuclease cleavage conformation”. We were already cautious in the first version of the manuscript which was stating “A conformation that we suggest corresponding to the endonuclease cleavage conformation...”. We updated the text and the figure,

being even more cautious “A conformation captured at 3.9 Å resolution shows the capped RNA bound to the CBD, 45 Å away from the ENDO, a distance compatible with capped RNA primer cleavage after 9 to 17 nucleotides (**Fig. 4c, d**). We called this snapshot the “putative endonuclease cleavage conformation”, although this remains speculative”

We also would like to point out that even if only two nucleotides of the capped RNA are visible, no structure with capped RNA visible from the CBD to the ENDO has ever been observed for any negative stranded RNA virus. In influenza virus polymerase, the conformation suggested to be compatible with cleavage by the ENDO does not display the capped RNA (Reich *et al.*, Nature, 2014, PDB: 4WSB, **Supplementary Fig. 13a**).

36. Line 239-240 — It would help to use consistent conformation names between the text and figures. No where is “capped primer active site entry” found in Fig 4.

The conformation names are consistent between the text and the figures. Indeed, we chose not to show the “capped primer active site entry” in **Fig. 4** because it would not bring new essential elements. Indeed, in the “capped primer active site entry” conformation, the protein, the 5'-1-17BPm, the 3'-vRNA1-25 (except U25) have the same conformation as in the “initiation” conformation, the visible part of the capped RNA is also in the same position as in the “initiation” conformation. We therefore think that adding it in **Fig. 4** would make the figure difficult to read and understand. Indeed, it would require adding a 5th structure for panel **c**, a 5th structure for panel **d** and a 5th structure for panel **e**, thereby crowding the figure, making each element too small.

We have added a sentence in the legend of **Fig. 4**: “Note that LACV-L in “capped primer active site entry” state has the same conformation as in “initiation” state.”

Please note that the “capped primer active site entry” is now shown in **Supplementary Fig. 9b**, that also answers Reviewer 1 request.

37. Line 245-247 — reword this sentence, it is difficult to read as is.

This sentence has been reworded: “Finally, another cryo-EM data collection and image processing led to the determination of a transcription “early-elongation” conformation structure at 3.3 Å resolution. It was obtained by incubating the transcription complex for 30 min at 30°C in presence of ATP, UTP, GTP and MgCl₂. It shows an elongated capped RNA that forms a 9-base pair template-product duplex in the active site cavity (**Fig. 4c, d, Supplementary Fig. 9d**).”

38. Line 254 “ENDO movement resulting in a 160” — please also label this in fig. 4e

The text and figure have been updated.

An arrow on the left panel of **Fig. 4e** has been added that indicates the 160° rotation.

39. Line 256 “In addition to the ENDO movement” — label thumb in fig. 5a

The text has been updated.

As there is no interaction between the CTER and the thumb, it does not appear logical to show the thumb. Reviewer 1 might have wanted to see interaction with the thumb ring, which is already shown.

40. Line 265 — It's hard to see this rotation in the figures themselves, might help to redesign the figures to highlight this movement.

Reviewer 1 refers here to the 175° rotation of the ENDO between the “endonuclease cleavage” conformation and the “transcription initiation” conformation. In this figure, the movement of the ENDO is clear, even though it is not seen in its rotation axis. We would like to pinpoint here in that, in **Fig. 4c, d and e**, the angle of view has been carefully chosen to visualize not only the movement of the ENDO, but also the localization of the ENDO cleavage site, the RNA path, the CBD positions in the different conformations and the rotation of the CTER. Each of the rotations, i.e. rotation of the ENDO, rotation of the CTER and rotation of the CBD, is large and has a different rotation axis, so it is not possible to see for each of them on its own rotation axis. However, to facilitate the visualization of the movements, the Supplementary Movies 1 and 2 have been done, they enable to see the structure in different views.

41. Line 281 — what makes cap binding peculiar?

The cap is stacked between a tryptophane (W1847) and an arginine (R1854). The stacking of a cap between these two amino acids had never been observed. This is interesting because during a long time it was impossible to decipher the CBD position based on sequence analysis. The presence of CBD in the polymerase was subject to controversy. We now understand why: each viral family of the *Bunyavirales* order adopts a different strategy, involving different amino acids (here a W and a R) to stack the capped RNA. These amino acids are conserved within a family. If requested by the Editor, we could specify this in the discussion. We haven't done it here as Reviewer 1 requests that we shorten the discussion (main point 4).

42. Line 293 “Because these residues bind to the phosphate backbone, it is likely that binding is sequence independent”

The text has been updated.

43. Line 296 “indicating tighter binding”

The text has been updated.

44. Line 300 — It is confusing what early-state initiation refers to.

To clarify, we have rephrased this sentence: “Following the transcription initiation state visualized in the structure”. It corresponds to an early initiation state because the realignment has not occurred yet in the state corresponding to the structure.

Reviewer #2 (Remarks to the Author):

Arragain et al have captured various elongating LACV RNA polymerase complexes. To obtain these structures, which had proved difficult so far, the authors changed the position of the tag on the polymerase, introduced a number of clever mutations in the promoter strands, and mutated the endonuclease domain of the RNA polymerase to prevent inadvertent cleavage of products.

One of the most exciting insights from these new structures is how prime-realignment may be achieved during negative strand virus replication and transcription. Prime-realignment is a process in which the RNA polymerase of negative strand RNA viruses copies 2-3 nucleotides once (or more) before starting processive elongation. This process has been observed for a number of RNA viruses, including the influenza A virus.

Through their new structures, Arragain et al link the prime-realignment mechanism to a novel structural element, which they call the PR loop. Mutation of this loop abrogates or increases RNA polymerase activity, underlining that the loop is important for polymerase activity. The authors also note that one of the mutations leads to the formation of products that migrate slower in denaturing PAGE, suggesting that there is a link between the function of the loop and the stimulation of prime-realignment. This part would be strengthened by sequencing data of e.g. RNA extracted from the mini-genome assays with the wt and mutant RNA polymerases.

Overall, I think that the data are presented with great clarity, that the conclusions are well-supported by the data, and that the insights from this study are important to the field. The data and insights are also in line with previous studies on the influenza A virus RNA polymerase, where mutation of a comparable loop also affected realignment.

I have a couple of points that would strengthen the authors' manuscript.

1. Have the realignment products from mini-replication assays been sequenced to confirm a) that the mutants are triggering more realignment events in cell culture and b) that the slower migrating bands identified as realignment products in PAGE are truly realignment products?

Reviewer 2 is referring to mini-replication assays in cell culture that were not present in the original article. We therefore assumed that Reviewer 2 wanted us to perform RNA sequencing on the *in vitro* mini-replication and transcription assays shown in **Fig. 2e** and **6c**. This analysis confirms that, for replication, the realignment events are more important for LACV-L_{H34KS991A} and LACV-L_{H34KI990A} than for LACV-L_{H34K}, in terms of percentage of products and in terms of read numbers.

It also confirms that the slower migrating bands correspond to replication products with 5' addition of GU or GGU sequences, added by realignment, as proposed in the initially submitted manuscript.

This information is now added on a newly incorporated **Supplementary Fig. 2a**. In addition, the text has been updated for replication assays:

"Slower-migrating products are also visible on gel and NGS identifies them as being products extended in 5' with GU and GGU sequences."

Concerning transcription, the comparison was done between LACV-L_{H34K} and LACV-L_{H34KI990A} that display different behavior on transcription assays. The NGS analysis shows that LACV-L_{H34KI990A} is triggering slightly more realignment than LACV-L_{H34K}, but more importantly, that LACV-L_{H34KI990A} is triggering the formation of many aberrant products that realign

repetitively, without finishing product formation. Altogether these results confirm the role of the PR loop in realignment.

The figures and main text have been updated to reflect the modifications:

-Supplementary Fig. 2 provides statistics about the NGS for transcription reactions.

-The text has been updated for transcription assays:

“The majority 40-/39-mer products correspond to primed and subsequently realigned transcripts, that result in the addition of a AGU nucleotide triplet at the end of the capped primer before elongation, as detected by NGS analysis (**Supplementary Fig. 2b**).”

-The text has been updated for transcription assays with PR loop mutants:

“While transcription assays performed with LACV-L_{H34KM989A} and LACV-L_{H34KS991A} are equivalent to LACV-L_{H34K}, LACV-L_{H34KI990A} displays a different transcription profile (**Fig. 6c**). NGS analysis reveals that LACV-L_{H34KI990A} generates, in addition to the expected 37- and 40-mer products, a significant number of aberrant products. These latter mainly correspond to the capped primer elongated by repetitive of AGU triplets that are likely added by realignment. Altogether these results confirm the role of the PR loop, and in particular of the residue 1990, in realignment during transcription activity.”

2. It would be helpful if the gel data in Fig. 2e were quantified and statistics performed.

The reactions corresponding to **Fig. 2e** were repeated three times. Quantification was performed in ImageJ for both the expected products and the GU/GGU 5' extended replication products. Two-tailed unpaired T-test are performed and are shown **Fig. 2e**. P-value are indicated and are considered significant when under 0.05. This shows the role of M989 in replication initiation, and identifies that the mutation 1990A and S991A significantly increase the formation of GU/GGU 5' extended replication products. The main text, **Fig 2e** legends and material and methods have been updated accordingly.

“The identification of the unexpected PR loop led us to analyze further its role in replication by engineering single-alanine substitutions of its tip residues M989, 1990 and S991 (**Fig. 2e**). M989A diminishes the formation of 25-mer product by 95% and of GU/GGU 5' extended replication products by 81%, confirming the importance of this residue in precise template positioning at initiation (**Fig. 2e, lane 2**). 1990A mutation does not significantly modify 25-mer product formation but multiplies by 3.9 the formation of GU/GGU 5' extended replication products. (**Fig. 2e, lane 3 vs lane 1**). LACV-L_{Citag_H34K_S991A} produces 10.7 times more GU/GGU 5' extended replication products than LACV-L_{Citag_H34K}. Altogether, these mutations clearly confirm the importance of the PR loop tip in the replication mechanism with a defined role of M989 at initiation and an unexpected role of the mutations 1990A and S991A in the formation of GU/GGU 5' extended replication products.”

Reviewers' Comments:

Reviewer #1:

Remarks to the Author:

The authors have address all my concerns.

Reviewer #2:

Remarks to the Author:

In this revised manuscript, Arragain et al describe structural and biochemical evidence that provide insight into the mechanism of LACV RNA polymerase elongation. Of particular interest is the proposed mechanism underlying prime-realignment, a process that is used by several RNA viruses. The revised manuscript has been considerably strengthened by the addition of new data, including next generation sequencing data, and textual clarifications. In my opinion, the biochemical and structural analyses are sound, and the conclusions thoroughly supported by the data and in line with previous observations in the field. I feel that my previous concerns have been adequately addressed.